# The antimicrobial peptide thanatin disrupts the bacterial outer membrane and inactivates the NDM-1 metallo-β-lactamase

Bo Ma[1,3], Chao Fang [1,3], Linshan Lu[2,3], Mingzhi Wang[1], Xiaoyan Xue[1], Ying Zhou[1], Mingkai Li[1], Yue Hu[1], Xiaoxing Luo[1] & Zheng Hou [1]

New Delhi metallo-β-lactamase-1 (NDM-1) is the most prevalent type of metallo-β-lactamase and hydrolyzes almost all clinically used β-lactam antibiotics. Here we show that the antimicrobial peptide thanatin disrupts the outer membrane of NDM-1-producing bacteria by competitively displacing divalent cations on the outer membrane and inducing the release of lipopolysaccharides. In addition, thanatin inhibits the enzymatic activity of NDM-1 by displacing zinc ions from the active site, and reverses carbapenem resistance in NDM-1-producing bacteria in vitro and in vivo. Thus, thanatin's dual mechanism of action may be useful for combating infections caused by NDM-1-producing pathogens.

[1] Department of Pharmacology, School of Pharmacy, Fourth Military Medical University, Xi'an 710032, China. [2] Department of Obstetrics and Gynecology, Tangdu Hospital, Fourth Military Medical University, Xi'an 710038, China. [3] These authors contributed equally: Bo Ma, Chao Fang, Linshan Lu. Correspondence and requests for materials should be addressed to X.L. (email: xxluo3@fmmu.edu.cn) or to Z.H. (email: hzh_0001@163.com)

New Delhi metallo-β-lactamase-1 (NDM-1)-producing bacteria have been rapidly developing resistance to almost all antibiotics, including carbapenems[1,2]. The extensive horizontal transfer of $bla_{NDM-1}$ among various types of Gram-negative strains, especially *Escherichia coli* (*E. coli*) and *Klebsiella pneumonia* (*K. pneumoniae*), accelerates the severe global spread of the gene[3,4]. In addition, $bla_{NDM-1}$ intercalates into existing resistance genes and evolutionarily generates new mutants. Previous studies reported that the resistance gene *mcr-1* can coexist with $bla_{NDM-1}$ in *E. coli*, thereby severely worsening such situation[5,6]. Considering the ever-increasing, fast-spreading and highly lethal NDM-1-producing strains, the development of new drugs is urgently needed to combat these pathogens.

The hydrolytic activity of NDM-1 depends on the binding of $Zn^{2+}$ ions to the active site, which activates nucleophilic $H_2O$ and causes the cleavage of β-lactam rings[7,8]. Nevertheless, in the absence of $Zn^{2+}$ ions, the stability of NDM-1 decreases significantly, which adversely affects the accumulation of this enzyme in the bacterial periplasm[9]. The innate immune system responds to metallo-β-lactamase-producing bacteria by releasing metal-chelating proteins, which will cause the degradation of metallo-β-lactamases[10,11]. However, resistant bacteria anchor NDM-1 to the outer membrane (OM) to avoid chelation so that the hydrolytic efficacy can be preserved even under the condition of metal depletion[9].

Thanatin is an inducible 21-residue insect peptide with a disulfide bond between Cys11 and Cys18[12]. Our previous studies revealed that thanatin exerts prominent antibacterial effects on extended-spectrum β-lactamase-producing *E. coli*[13]. However, whether thanatin is also effective against NDM-1-producing bacteria remains unclear. In the present study, we examined the antibacterial efficacy of thanatin on NDM-1-producing strains and found that it affects both bacterial viability and NDM-1 enzyme activity. Our results showed that thanatin has the property of competitive replacement of divalent cations from bacterial OM, leading to OM disruption. Remarkably, thanatin also acts as an antibiotic adjuvant even under low concentrations by displacing zinc ions from the active site of the NDM-1 enzyme to restore the susceptibility of NDM-1-producing pathogens to β-lactam antibiotics. The unique antibacterial mechanisms of thanatin provide a strategy for combating infections of NDM-1 pathogens.

## Results

**Thanatin exerts bactericidal activity by disrupting OM integrity.** Seven clinically isolated NDM-1-producing strains and three

reference strains were used to assess the in vitro antibacterial activity of thanatin. NDM-1 expression was verified in all seven clinically isolated strains (Supplementary Fig. 1). As predicted, thanatin exhibited potent inhibitory effect on the growth of all NDM-1-producing *E. coli* and *K. pneumoniae* strains at 0.4–3.2 μM of the minimum inhibitory concentration (MIC) values (Table 1), whereas these strains showed much lower susceptibilities to penicillin, cephalosporin, carbapenem, and quinolone antibiotics (Table 1, Supplementary Tables 1 and 2). The bactericidal kinetics study of thanatin at the MIC levels showed rapid bacterial reduction from $10^6$ to $10^2$ and less than 50 colony-forming units (CFU)/mL at 1 and 3 h, respectively (Fig. 1a, b, Supplementary Fig. 2). The OM integrity was damaged by thanatin in a time-dependent manner, and a distinct increase in the fluorescence intensity was observed 1 h post incubation (Fig. 1c). The fluorescence intensity of propidium iodide (PI) increased ~2 h post incubation, indicating an increase in the inner membrane permeability of the bacteria (Fig. 1d). The scanning electron microscopy image showed that the number of *E. coli* cells with dramatic morphological changes gradually increased with increasing thanatin concentration; consequently, the cells exhibited aggravating corrugation on the surface (Fig. 1e).

**Thanatin damages the OM by promoting the release of divalent cations.** Divalent cations, which link the negatively charged phosphate groups between lipopolysaccharides (LPS) molecules via ionic bridges, are crucial for the OM integrity of Gram-negative bacteria[14]. The replacement or chelation of divalent cations would disrupt OM integrity and permeabilize bacteria. The release of $Ca^{2+}$ and LPS was detected to further investigate the effects of thanatin on the OMs of NDM-1-producing bacteria. NDM-1-producing *E. coli* XJ141026 showed an immediate release of $Ca^{2+}$ into the supernatant after incubation with thanatin at concentrations of 13 or 26 μM as early as 0.5 h post incubation (Fig. 2a). Thereafter, LPS was released at 3 h, indicating the gradual destruction of OM under the condition of divalent cation loss (Fig. 2b). Consistently, the bacterial loads of the thanatin-treated groups significantly decreased 3 h post incubation (Fig. 2c).

We then determined the release of $Ca^{2+}$ and LPS induced by thanatin from bacterial OM in a mouse pneumonia model infected with NDM-1-producing *E. coli* XJ141026 or *K. pneumoniae* XJ155017. The results showed that the LPS concentration in bronchoalveolar lavage fluid (BALF) increased

## Table 1 MICs of thanatin and antibiotics in seven NDM-1-producing strains

| Strains | MICs | | | | | | |
|---|---|---|---|---|---|---|---|
| | Thanatin (μg/mL)/(μM) | Piperacillin (μg/mL) | Oxacillin (μg/mL) | Meropenem (μg/mL) | Imipenem (μg/mL) | Ceftazidime (μg/mL) | Cefotaxime (μg/mL) |
| *E. coli* ATCC25922 | 4/1.6 | 2 | 64 | <0.125 | <0.125 | <0.125 | <0.125 |
| *E. coli* ATCC35218 | 1/0.4 | >256 | 256 | <0.125 | <0.125 | <0.125 | <0.125 |
| *K. pneumoniae* ATCC13883 | 4/1.6 | 8 | 64 | <0.125 | <0.125 | <0.125 | <0.125 |
| NDM-1 *E. coli* XJ141015 | 2/0.8 | >256 | >256 | 64 | 16 | >256 | >256 |
| NDM-1 *E. coli* XJ141026 | 2/0.8 | >256 | >256 | 64 | 16 | >256 | >256 |
| NDM-1 *E. coli* XJ141047 | 2/0.8 | >256 | >256 | 64 | 32 | >256 | >256 |
| NDM-1 *K. pneumoniae* XJ155017 | 8/3.2 | >256 | >256 | >256 | >256 | >256 | >256 |
| NDM-1 *K. pneumoniae* XJ155018 | 8/3.2 | >256 | >256 | 128 | 32 | >256 | >256 |
| NDM-1 *K. pneumoniae* XJ155019 | 8/3.2 | >256 | >256 | 128 | 128 | >256 | >256 |
| NDM-1 *K. pneumoniae* XJ155020 | 8/3.2 | >256 | >256 | 128 | 128 | >256 | >256 |

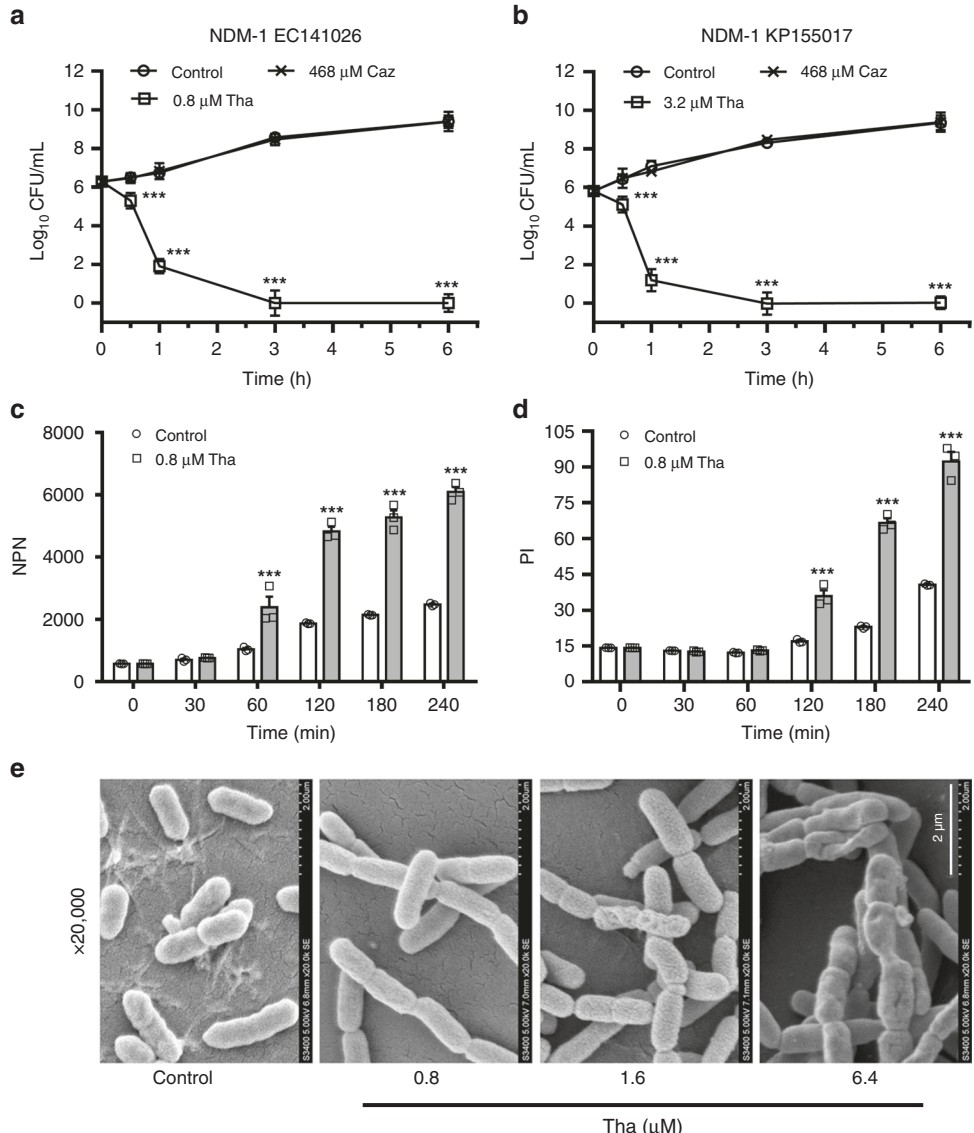

**Fig. 1** Thanatin disrupts the integrity of bacterial membrane. **a, b** Kill curves of thanatin (Tha) and ceftazidime (Caz) against NDM-1-producing *E. coli* XJ141026 (**a**) and *K. pneumoniae* XJ155017 (**b**). Cell numbers were determined by plating for colony counts. **c, d** Outer and inner membrane permeabilization of thanatin was measured by detecting the fluorescence intensity of NPN (**c**) and PI (**d**) in *E. coli* XJ141026. **e** Morphology of *E. coli* XJ141026 was investigated by scanning electron microscopy at 4 h after thanatin treatment. Scale bar = 2 μm. All data are shown as the mean ± s.e.m. from three independent experiments. *P*-values were determined by two-way ANOVA; \*\*\**P* < 0.001 vs. control. Source data are provided in Source Data file

significantly after treatment with thanatin for 1 h, whereas no significant difference was observed at 6 h (Fig. 2d, Supplementary Fig. 3a). The lack of increase in LPS release in BALF at 6 h may be associated with the decrease in the bacterial titers after thanatin treatment (Fig. 2e, Supplementary Fig. 3b). The survival rates were improved from 40 to 90% after treatment with 6 mg/kg thanatin in *E. coli* XJ141026-infected mice (Fig. 2f) and from 50 to 100% after treatment with 9 mg/kg thanatin in *K. pneumoniae*-infected mice (Supplementary Fig. 3c). The increasing survival rates caused by thanatin treatment were associated with reduced bacterial titers and restricted pneumonia aggravation in the lungs of the infected mice (Supplementary Figs. 3d, e and Supplementary Figs. 4a, b).

**Thanatin competitively replaces divalent cations from LPS.** To verify the mechanism of thanatin in promoting the release of LPS and Ca²⁺, we investigated the relationship among divalent cations, LPS, and antibacterial activity of thanatin. Our results showed that thanatin-induced LPS release from NDM-1-producing *E. coli* XJ141026 was significantly reduced with increasing Ca²⁺ concentrations (Fig. 3a). Meanwhile, the bactericidal efficacy of thanatin was inhibited by adding extra divalent ions into the systems (Fig. 3b). The MICs increased dramatically with increasing concentrations of Ca²⁺ and Mg²⁺ (Fig. 3c). Consistently, the chelation of the divalent cations by metal-chelating agent dipicolinic acid (DPA) and ethylenediaminetetraacetic acid (EDTA) facilitated the bactericidal activity of thanatin, that is, the MICs were decreased to 0.0625- and 0.1563-fold compared with that when thanatin alone was used (Fig. 3d). These results indicated that thanatin likely competitively replaces the divalent cations from OM to exert antibacterial activity. To acquire direct evidence, we examined the affinities of thanatin, Ca²⁺, and Mg²⁺ to LPS by isothermal titration calorimetry (ITC). The equilibrium dissociation constant ($K_d$) of thanatin to LPS was 1.09 ± 0.11 μM, whereas the values for Ca²⁺ and Mg²⁺ were

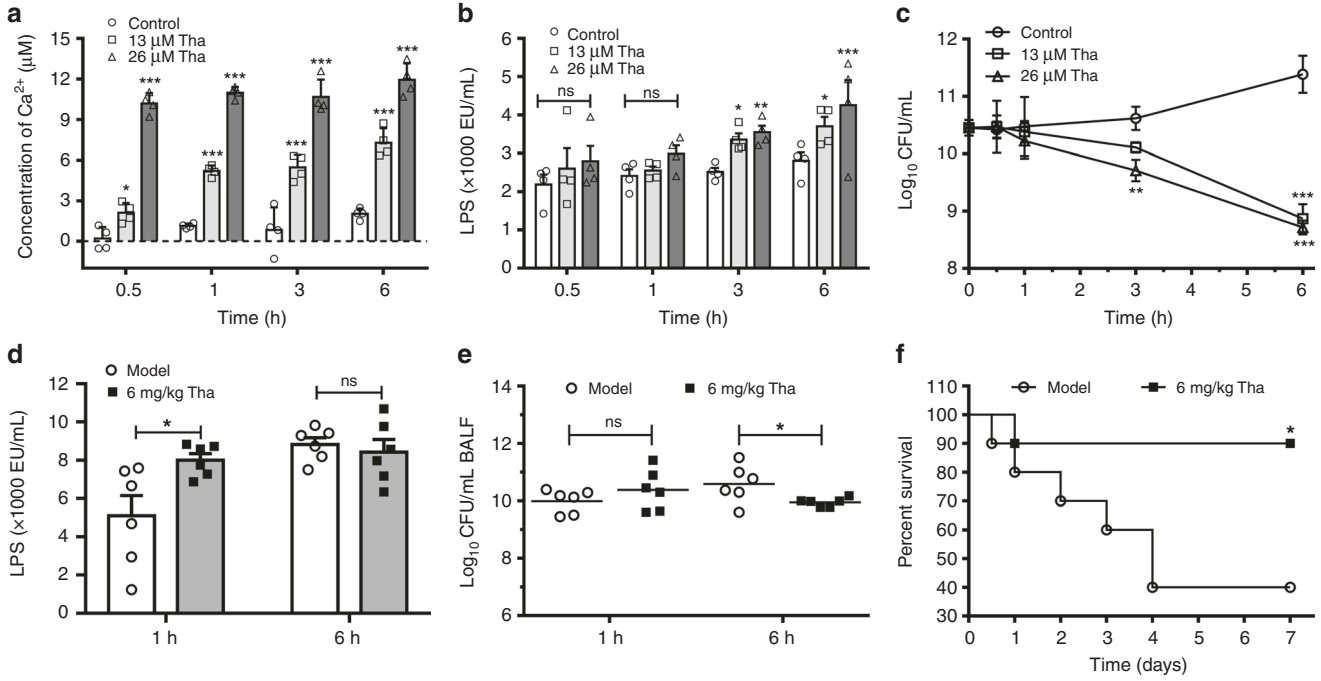

**Fig. 2** Thanatin promotes the release of $Ca^{2+}$ and LPS from the OM. **a–c** Release of $Ca^{2+}$ (**a**) and LPS (**b**) from thanatin-treated *E. coli* XJ141026 ($3 \times 10^{10}$ CFU/mL) was detected, and time-kill curves were monitored (**c**). **d, e** LPS levels (**d**) and bacterial loads (**e**) in the BALF of *E. coli* XJ141026-infected pneumonia mice were analyzed at 1 and 6 h after 6 mg/kg thanatin treatment (*n* = 6 per group). **f** Survival curves for the *E. coli* XJ141026 pneumonia model. BALB/c mice were intranasally infected with a sublethal dose of *E. coli* XJ141026 and treated with 6 mg/kg thanatin via intraperitoneal injection (*n* = 10 per group). All data are shown as the mean ± s.e.m. from at least three independent experiments. *P*-values were determined by two-way ANOVA (**a–c**), two-tailed unpaired *t*-test (**d, e**) or log-rank test (**f**). ns means not significant; \**P* < 0.05, \*\**P* < 0.01, \*\*\**P* < 0.001 vs. control or model. Source data are provided in Source Data file

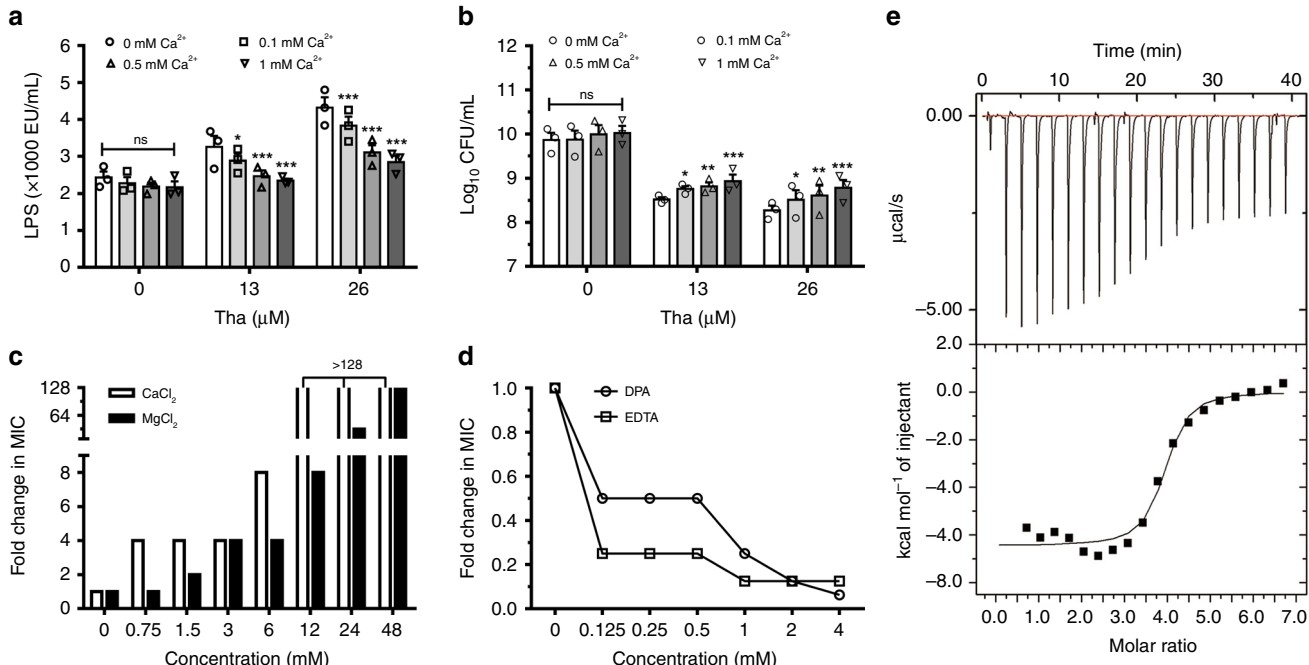

**Fig. 3** Thanatin replaces divalent cations from LPS. **a, b** Effects of $Ca^{2+}$ on thanatin-induced LPS release from NDM-1-producing *E. coli* XJ141026 (**a**) and CFUs (**b**) after treatment with 13 or 26 μM thanatin. **c, d** Effect of the divalent cations $Ca^{2+}$ or $Mg^{2+}$ (**c**) or the metal-chelating agents DPA or EDTA (**d**) on the MIC of thanatin against NDM-1-producing *E. coli* XJ141026. **e** ITC thermograms for the binding of LPS to thanatin. The downward peaks indicate an exothermic process. All data are shown as the mean ± s.e.m. from three independent experiments. *P*-values were determined by two-way ANOVA (**a, b**). ns means not significant; \**P* < 0.05, \**P* < 0.01, \*\*\**P* < 0.001 vs. control (0 mM $Ca^{2+}$). Source data are provided in Source Data file

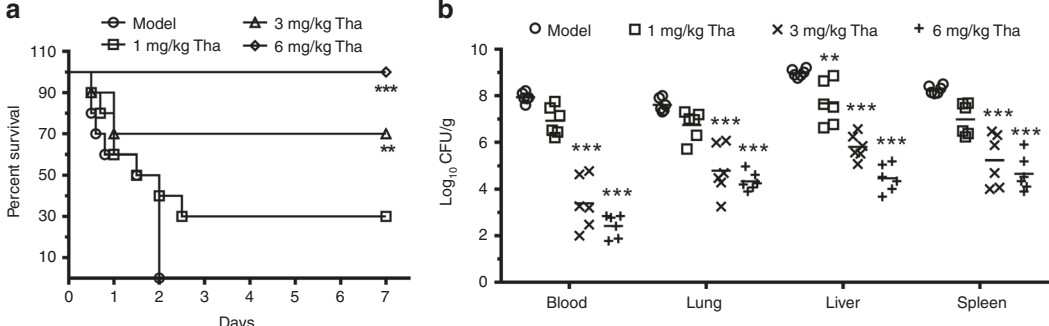

**Fig. 4** Thanatin protects NDM-1-producing *E. coli*-infected mice. **a** Survival curves for the *E. coli* XJ141026 sepsis model. BALB/c mice were intraperitoneally administered with a lethal dose of *E. coli* XJ141026 and treated with three doses of thanatin via intraperitoneal injection ($n = 10$ per group). **b** Bacterial loads in the blood, lungs, livers, and spleens of thanatin-treated *E. coli* XJ141026-infected mice were determined 24 h after infection ($n = 6$ per group). P-values were determined by log-rank test (**a**) or one-way ANOVA with post-Bonferroni's comparison test (**b**). **$P < 0.01$, ***$P < 0.001$ vs. model. Source data are provided in Source Data file

much higher, indicating the stronger affinity of thanatin than that of $Ca^{2+}$ or $Mg^{2+}$ to LPS (Fig. 3e, Supplementary Fig. 5 and Supplementary Table 3). All these data suggested that thanatin competitively replaces the divalent cations to bind to LPS, thereby disrupting the integrity of OM and leading to bacterial death.

**Thanatin protects mice infected with NDM-1-producing *E. coli*.** We examined the in vivo therapeutic effects of thanatin in a sepsis model induced by NDM-1-producing *E. coli* XJ141026. All septic mice died within 2 d post infection, whereas treatment with 1, 3, and 6 mg/kg thanatin markedly increased the survival rate from 0 to 30%, 70%, and 100%, respectively (Fig. 4a). Tissues were collected 24 h post infection to analyze bacterial CFUs after treatment. The results showed that the bacterial titers decreased with increasing thanatin dose (Fig. 4b). The therapeutic effects were confirmed by histological staining. We observed dramatic pathological changes in the model group, including large amounts of inflammatory cell infiltration, alveolar fusion, congestion in the spleen red pulp area, hepatic sinusoidal dilation and congestion (Supplementary Fig. 6). By contrast, the thanatin treatment rescued the pathological damages in a dose-dependent manner (Supplementary Fig. 6).

To examine the cell toxicity of thanatin, we used human umbilical vein endothelial cells (HUVECs), human pulmonary alveolar epithelial cells (HPAEpiCs), and mouse neuron cells. At the concentration of 200 μM, almost all thanatin-treated HPAEpiCs were alive, but only 73.8% of cells survived after colistin treatment (Supplementary Fig. 7a). Similarly, 82.9% of thanatin-treated HUVECs survived, but 60.5% of colistin-treated cells were alive at the same concentration of 200 μM (Supplementary Fig. 7b). These results indicated that thanatin had lower toxicity toward HPAEpiCs and HUVECs than colistin. Hoechst 33342 and PI staining assay results showed that no differences between the control and 50 μM thanatin-treated groups when incubated with mouse primary neuron cells (Supplementary Fig. 7c). This finding implied that thanatin exhibited high selectivity toward bacterial cell walls but not on mammalian cell membranes.

**Thanatin inactivates NDM-1 by displacing zinc ions.** The NDM-1 protein is anchored to the OM of Gram-negative bacteria and is involved in enzymatic reaction in the periplasmic space[9]. Both thanatin and colistin damage the OM integrity, causing the release of NDM-1 into the solution in a time- and concentration-dependent manner (Fig. 5a, Supplementary Figs. 8a, b and 13).

Meanwhile, the levels of NDM-1 decreased in the thanatin- and colistin-treated *E. coli* XJ141026 cell precipitates (Supplementary Figs. 8b, c and 13). As a result, the hydrolysis rates of the precipitates consistently decreased with increasing thanatin or colistin concentration due to the release of NDM-1 into the supernatant (Supplementary Figs. 8d, e). We speculated that high NDM-1 levels in the supernatant were related to the efficient hydrolytic activity. Unexpectedly, the hydrolysis of the supernatant to imipenem decreased with increasing thanatin concentration (Fig. 5b). By contrast, treatment with high colistin concentration led to strong hydrolytic activity to imipenem (Supplementary Fig. 8f).

In addition to membrane permeabilization, thanatin likely interacted with NDM-1 and inhibited its hydrolytic activity. To determine the direct interaction between NDM-1 and thanatin, microscale thermophoresis (MST) was used to detect the binding affinities of apo-NDM-1 to $Zn^{2+}$, thanatin and colistin. We measured a $K_d$ of $\sim 0.71 \pm 0.06$ μM for the interaction between NDM-1 and thanatin (Fig. 5c). Weak affinities to NDM-1 were detected with $K_d$ values of $7.36 \pm 0.45$ μM for $Zn^{2+}$ and $61.68 \pm 4.92$ μM for colistin (Supplementary Figs. 9a, b). These results suggested that thanatin exhibited approximately 10 times higher affinity to NDM-1 than $Zn^{2+}$.

To further define whether thanatin can inhibit the enzymatic activity of NDM-1, we incubated the purified NDM-1 protein with gradient concentrations of thanatin, and examined the hydrolytic efficacy. The results showed that thanatin inhibited NDM-1 activity in a concentration-dependent manner, with the half-maximum inhibitory concentration ($IC_{50}$) value of $3.21 \pm 0.78$ μM (Fig. 5d). Classical kinetic plots with varied substrate and inhibitor concentrations were determined to show the manner by which thanatin inhibited NDM-1. The relevant Lineweaver–Burk plot proved that thanatin was a competitive inhibitor for NDM-1, with the inhibition constant ($K_i$) value of $2.84 \pm 0.33$ μM (Fig. 5e, Supplementary Fig. 10). The kinetic parameters are shown in Supplementary Table 4. These results suggested that thanatin could directly inhibit enzymatic activity of NDM-1 as a competitive inhibitor.

NDM-1 is a dizinc hydrolase with two $Zn^{2+}$ ions in its active site[15,16]. To directly confirm whether thanatin inhibits NDM-1 by removing $Zn^{2+}$ from NDM-1, we used inductively coupled plasma–mass spectrometry (ICP-MS) to measure the zinc content in holo-NDM-1 after thanatin treatment. The results confirmed that $2.23 \pm 0.28$ molar equivalents of zinc were bound to holo-NDM-1 (Fig. 5f). However, only $1.03 \pm 0.25$ molar equivalents of zinc remain bound to NDM-1 after thanatin treatment (Fig. 5f).

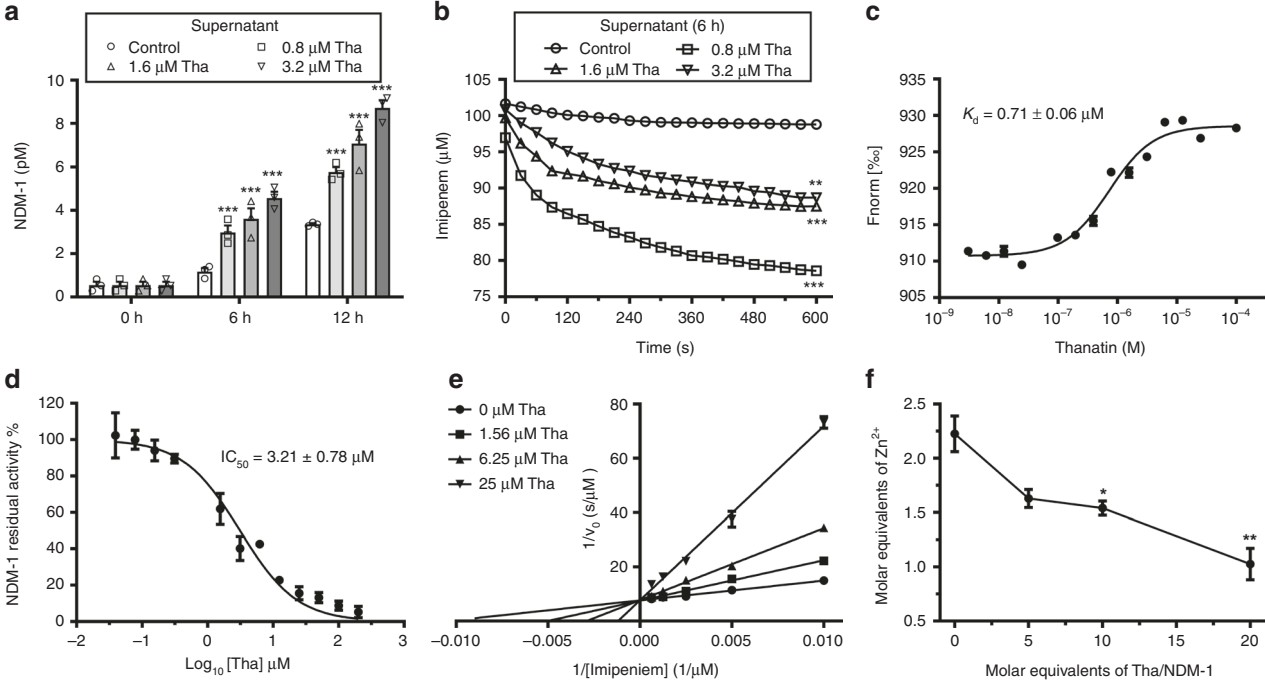

**Fig. 5** Thanatin inactivates NDM-1. **a** Thanatin induced NDM-1 release to *E. coli* culture supernatant. **b** Hydrolytic effects of the supernatant on imipenem. The supernatant was obtained from thanatin-treated NDM-1-producing *E. coli* XJ141026 at 6 h. **c** $K_d$ value for binding of thanatin to RED-tris-NTA-labeled apo-NDM-1 was obtained by MST assay. **d** Thanatin inhibited the activity of NDM-1. **e** The Lineweaver–Burk plot. Thanatin is a competitive inhibitor of NDM-1. **f** Molar equivalents of zinc in holo-NDM-1 were detected by ICP-MS after thanatin treatment. Data are shown as the mean ± s.e.m. from three independent experiments. *P*-values were determined by two-way ANOVA (**a**, **b**), or one-way ANOVA with Bonferroni's comparison test (**f**). *$P < 0.05$ **$P < 0.01$, ***$P < 0.001$ vs. control. Source data are provided in Source Data file

Hence, thanatin released ~1.2 molar equivalents of zinc from holo-NDM-1. In addition, NDM-1 activity inhibited by thanatin was gradually rescued with increasing $Zn^{2+}$ concentrations (Supplementary Fig. 11). Hence, thanatin inhibited NDM-1 reversibly by removing $Zn^{2+}$ from NDM-1. Thanatin, as a cationic peptide, bears a significant positive charge ($+6$) at physiological pH[12], implying a repulsive force against metal ions. Although cysteine residues can bind to $Zn^{2+}$[17], the two cysteines in thanatin have formed a disulfide bridge and cannot function as metal-chelating residues. Therefore, thanatin likely inactivated NDM-1 by displacing the $Zn^{2+}$ ions, rather than by exerting a chelating effect. All the data showed that thanatin inactivated NDM-1 by displacing $Zn^{2+}$ as a competitive inhibitor.

**Thanatin reverses carbapenem resistance in vivo**. The capacity of NDM-1 inhibition suggested that thanatin could potentially protect conventional antibiotics from hydrolysis and restore the antibiotic susceptibility of NDM-1-producing strains. The results of the checkerboard assay showed that, in the presence of sub-MIC thanatin, the antibacterial activity of meropenem toward NDM-1-producing *E. coli* XJ141026 was markedly restored, with the MIC value decreased from 144 μM to 18 μM (Fig. 6a). All fractional inhibitory concentration (FIC) indices were ≤1 for the seven NDM-1-producing strains, indicating that thanatin exerted synergistic or additive effects with imipenem and meropenem (Table 2). The combination of meropenem/imipenem and thanatin showed great bactericidal capacities against NDM-1-producing *E. coli* XJ141026 at concentrations much lower than MICs when monoadministered (Fig. 6b, c). The effects of the combination treatment were determined with the six other NDM-1-producing strains. Consistent results were obtained, indicating that thanatin overcame the antibiotic resistance by inhibiting the

hydrolytic activity of NDM-1 (Supplementary Fig. 12). In the systemic infection model, the combination of meropenem and thanatin enhanced the survival rates from 0% for the 0.1 mg/kg thanatin monotherapy group and 30% for the 10 mg/kg meropenem monotherapy group to 79% (Fig. 6d). Bacterial titer data showed that the combination of meropenem and thanatin efficiently restricted the development of sepsis, and the CFUs in the spleen and liver were significantly reduced compared with any other monotherapy groups (Fig. 6e, f).

## Discussion

Efficient antibiotic treatment must be developed for serious infections caused by NDM-1-producing super-resistant bacteria. Current therapeutic strategies by using polymyxins and tigecycline are problematic due to their high toxicity, poor tissue penetrability, and undesirable pharmacokinetic properties[18–20]. Recent studies have reported that several compounds exert potent activities against NDM-1-producing bacteria[21,22]. However, the situation of restraining infections caused by NDM-1-producing strains is still severe, given high resistance to almost all β-lactam antibiotics, especially carbapenems. Here, we propose thanatin as a promising agent to combat NDM-1-producing bacteria by permeabilization and NDM-1 enzyme inactivation. Thanatin may be potentially used as a therapeutic strategy.

Divalent cations ($Mg^{2+}$ and $Ca^{2+}$) are essential to bridge negative-charged phosphate groups between the LPS molecules, avoiding the accumulation of repulsive forces and maintaining the stability of the bacterial OM[14]. Our results indicated that thanatin causes the release of $Ca^{2+}$ and LPS in a concentration-dependent manner (Fig. 2a, b). The same phenomenon was also observed in vivo, that is, the LPS levels in BALF significantly increased in the mouse pneumonitis model after thanatin

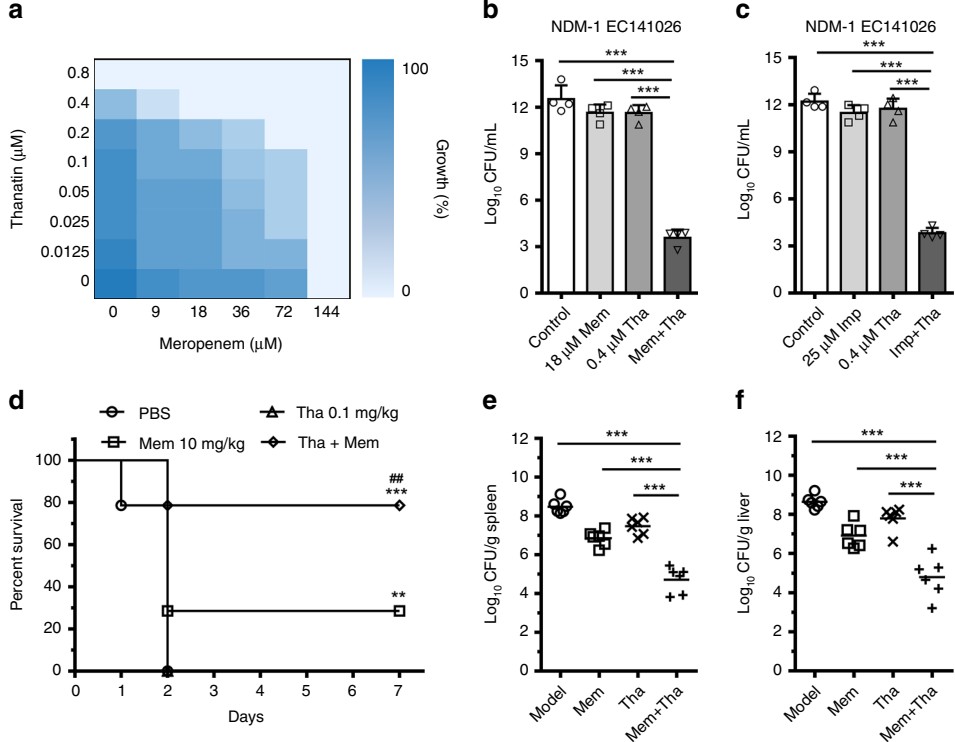

**Fig. 6** Thanatin rescues the activity of carbapenem. **a** Microdilution checkerboard analysis showing the combined effect of thanatin (Tha) and meropenem (Mem) against NDM-1-producing *E. coli* XJ141026. The heat plot showed an average of three technical replicates. **b**, **c** Sub-MICs of thanatin reversed the activity of meropenem (**b**) and imipenem (Imp) (**c**) resistance in vitro. **d** NDM-1-producing *E. coli* XJ141026-infected mice were treated with a single dose of meropenem (10 mg/kg), a combination of meropenem (10 mg/kg) and thanatin (0.1 mg/kg), thanatin alone (0.1 mg/kg), or PBS via intraperitoneal injection (*n* = 14 per group). **e**, **f** Bacterial loads in the spleen (**e**) and liver (**f**) of thanatin-treated *E. coli* XJ141026-infected mice were determined by plating the samples for colony counts 24 h after treatment (*n* = 6 per group). Data are shown as the mean ± s.e.m. from at least three independent experiments. *P*-values were determined by one-way ANOVA with Bonferroni's comparison test (**b**, **c**, **e**, **f**), or by log-rank test (**d**). \*\**P* < 0.01, \*\*\**P* < 0.001; ##*P* < 0.01 between 10 mg/kg Mem and Tha + Mem. Source data are provided in Source Data file

| Table 2 FIC indices against NDM-1 clinical isolates | | |
|---|---|---|
| **Strains** | **FIC Index** | |
| | **Mem + Tha** | **Imp + Tha** |
| NDM-1 *E. coli* XJ141015 | 0.625 | 1 |
| NDM-1 *E. coli* XJ141026 | 0.625 | 1 |
| NDM-1 *E. coli* XJ141047 | 0.508 | 0.531 |
| NDM-1 *K. pneumoniae* XJ155017 | 0.750 | 0.750 |
| NDM-1 *K. pneumoniae* XJ155018 | 0.563 | 1 |
| NDM-1 *K. pneumoniae* XJ155019 | 0.516 | 0.313 |
| NDM-1 *K. pneumoniae* XJ155020 | 0.508 | 0.375 |
| *Tha* thanatin, *Mem* meropenem, *Imp* imipenem | | |

treatment for 1 h (Fig. 2d, Supplementary Fig. 3a). These effects can be significantly inhibited by increasing the concentrations of divalent cations; as a result, the antibacterial efficacy of thanatin was adversely affected (Fig. 3). The affinity of thanatin is much higher than that of $Ca^{2+}$ or $Mg^{2+}$ to LPS (Fig. 3e, Supplementary Fig. 5), suggesting that thanatin competitively replaces the divalent cations from OM and kills Gram-negative bacteria by permeabilization. Sinha et al.[23] found that thanatin forms an antiparallel β-sheet structure with the LPS micelle through solution nuclear magnetic resonance; such structure, increases the hydrophobicity and cationicity of thanatin to LPS. These findings provide evidence to support the hypothesis that thanatin plays an antibacterial role by replacing divalent cations. Vetterli et al.[24]

reported that thanatin targets LptA and LptD in the network of periplasmic protein–protein interactions, leading to the inhibition of LPS transport and OM biogenesis in *E. coli*. However, they proposed that thanatin has no membrane-permeabilizing effect, which contradicts the present results. We consider that the distinct data raised by Vetterli et al. may be due to the high concentrations of $Ca^{2+}$ (1 mM) and $Mg^{2+}$ (0.5 mM) used in the membrane permeabilization assays. According to our results, the antibacterial activity of thanatin obviously decreased under the effect of 1 mM $Ca^{2+}$; that is, the MIC of thanatin against NDM-1-producing *E. coli* XJ141026 increased four times (Fig. 3c).

NDM-1 inhibitors, which antagonize multiple subtypes of metallo-β-lactamases and protect β-lactam antibiotics from being hydrolyzed, have been widely studied for synergistic application with β-lactam antibiotics to restore their bactericidal effects[15,25,26]. The most common mechanism of metallo-β-lactamase inhibitors is metal ion binding[27]. A recent study showed that the naturally occurring fungal compound aspergillomarasmine A inhibits the NDM-1 protease activity by chelating zinc ions from the active site[28]. Distinct from the direct metal ion binding, our results demonstrated that thanatin acts as an NDM-1 inhibitor by competitively displacing zinc ions. This effect was only observed in thanatin; whereas colistin exhibited almost no effect on NDM-1 activity, indicating that not all cationic antimicrobial peptides can affect enzymolysis (Fig. 5, Supplementary Figs. 8 and 13). This property indicates that thanatin can be used as an NDM-1 inhibitor to effectively restore the susceptibility of NDM-1-producing bacteria to conventional antibiotics (Fig. 7).

**Fig. 7** Mechanism of action of thanatin. Thanatin disrupts outer membrane integrity and overcomes metallo-β-lactamase-mediated antibiotic resistance by competitively displacing divalent cations

The survival of eukaryotes (HUVECs, HPAEpiCs, and mouse neuron cells) was barely affected after incubation with 100 μM thanatin (Supplementary Fig. 7), which is much higher than the antibacterial concentration of thanatin (MIC = 0.4–3.2 μM). Compared with the broadly used polypeptide antibiotic colistin, thanatin exhibits not only excellent antibacterial efficacy but also low cytotoxicity. As such, thanatin is a safe antibacterial agent when applied in vivo and has the potential to combat infections caused by NDM-1-producing bacteria.

In summary, we propose that the antibacterial activity of thanatin is due to dual effects on both the bacterial OM and the NDM-1 enzyme. Thanatin kills NDM-1-producing bacteria and efficiently increases the survival rate of infected mice without conferring obvious toxicity. As such, thanatin is a promising candidate to combat the emergence and dissemination of NDM-1-producing bacteria.

## Methods

**Bacterial strains**. *E. coli* ATCC25922, *E. coli* ATCC35218, and *K. pneumonia* ATCC13883 were used as references based on the Chinese National Center for Surveillance of Antimicrobial Resistance. The clinical isolates of *E. coli* XJ141015, *E. coli* XJ141026, *E. coli* XJ141047, *K. pneumonia* XJ155017, *K. pneumonia* XJ155018, *K. pneumonia* XJ155019, and *K. pneumonia* XJ155020 were obtained from the clinical laboratory of Xijing Hospital (Xi'an, China).

**Screening and identification of NDM-1-producing strains**. Clinical isolate strains were identified and screened by the Phoenix 100 automated bacterial identification system to rapidly detect antimicrobial resistance. Total DNA was extracted by the TIANamp Bacteria DNA Kit according to the manufacturer's protocol. The presence of $bla_{NDM-1}$ was confirmed using a diagnostic kit for NDM-1 (Puruikang Bio Inc., Shenzhen, China) based on fluorescent quantitative polymerase chain reaction assay. According to the instruction, Ct values of less than 37 were considered positive.

**Synthesis of thanatin**. Thanatin (GSKKPVPIIYCNRRTGKCQRM) was synthesized by the solid-phase method applying Fmoc (9-fluorenylmethyloxycarbonyl) active ester chemistry[12,29]. The crude peptide was purified to over 98% chromatographic homogeneity by reverse-phase high-performance liquid chromatography and confirmed by mass spectrometry analysis.

**Expression and purification of NDM-1**. The $bla_{NDM-1}$ gene was synthesized by Integrated DNA Technologies (Detai Bio-Tech Co., Ltd., Nanjing, China) by deleting the signal peptide gene sequence and introducing a 6 × His gene sequence at the 5′-site. The gene was then cloned into pET30a digested with NdeI and HindIII. The vector was used to encode the NDM-1 enzyme fused to the amino-terminal histidine tag (MHHHHHHMPGEIRPTIGQQMETGDQRFGDLVFRQ LAPNVWQHTSYLDMPGFGAVASNGLIVRDGGRVLVVDTAWTDDQTAQILN WIKQEINLPVALAVVTHAHQDKMGGMDALHAAGIATYANALSNQLAPQEG MVAAQHSLTFAANGWVEPATAPNFGPLKVFYPGPGHTSDNITVGIDGTDIA FGGCLIKDSKAKSLGNLGDADTEHYAASARAFGAAFPKASMIVMSHSAPDSR AAITHTARMADKLR).

For protein production, pET30a-NDM-1 was transferred into *E. coli* BL21 (DE3), and the cells were grown in Luria-Bertani (LB) medium containing 50 μg/mL kanamycin. A single colony of *E. coli* BL21 (DE3)/pET30a-NDM-1 was inoculated into 100 mL of LB culture containing 50 μg/mL kanamycin and grown overnight at 37 °C. The overnight culture was diluted in LB containing 50 μg/mL

kanamycin and grown to midlogarithmic phase (OD$_{600}$ = 0.6). Protein expression was induced by adding 0.5 mM isopropyl-β-D-thiogalactopyranoside (IPTG) and 0.5 mM ZnSO$_4$ to the medium. After overnight induction at 15 °C for 16 h, the cells were harvested by centrifugation at 8000 rpm for 15 min at 4 °C and then resuspended in buffer A (50 mM Tris-HCl buffer, pH = 8.0) containing 300 mM NaCl, 20 mM imidazole, 1% Triton X-100, 1 mM DTT, and 1 mM PMSF. After sonication, the mixture was centrifuged at 12,500 rpm for 15 min at 4 °C. The supernatant was filtered through a 0.45 μm filter (Millipore, USA), loaded onto a Ni-IDA column, and pre-equilibrated with buffer A. The column was washed with buffer A, and the protein was eluted by a linear gradient of imidazole from 50 mM to 500 mM in buffer A. The soluble NDM-1 protein was dialyzed against 2 L of 10 mM PBS (pH = 7.4) overnight at 4 °C and stored at −80 °C. The purified enzyme was verified to be >90% pure as assessed by SDS-PAGE. Protein concentration in the solution was measured with a Bradford protein assay kit, with bovine serum albumin as the standard. Zinc content in the purified NDM-1 was determined using the inductively coupled plasma–mass spectrometry (ICP-MS), ranging from 1.0 to 1.3 molar equivalents of zinc ion.

**Preparation of apo-NDM-1**. The apo form of NDM-1 was prepared by dialysis against EDTA as described by González et al.[9]. Briefly, apo-NDM-1 was derived from the purified NDM-1 obtained above by the two rounds of dialysis against 100 volumes of 10 mM HEPES, 200 mM NaCl, and 20 mM EDTA at pH = 7.4 over a 12 h period under stirring. EDTA was removed from the resulting apoenzyme solution by three dialysis steps against 200 volumes of 10 mM HEPES, 1 M NaCl, pH = 7.4, Chelex 100 and finally three dialysis steps against 200 volumes of 50 mM HEPES, 200 mM NaCl, pH = 7.4, and Chelex 100. All buffer solutions used to prepare the apoenzymes were treated by stirring with Chelex 100 (Bio-Rad). Zinc content in the apoprotein samples was checked using ICP-MS, and apo-NDM-1 contains less than 0.2 molar equivalents of zinc ion.

**Minimum inhibitory concentration (MIC)**. MIC values were identified through microdilution in sterilized 96-well polypropylene microtiter plates according to the broth microdilution guideline of the Clinical and Laboratory Standards Institute[30]. The test medium was Mueller–Hinton broth (MHB), and the strain concentrations were adjusted to $5 \times 10^5$ CFU/mL. After 16 h of incubation at various concentrations of thanatin or other drugs at 37 °C, MIC was defined as the lowest concentration of antibiotic with no visible growth.

The MHB medium was supplemented with varying concentrations of MgCl$_2$, CaCl$_2$, DPA, or EDTA to explore the effects of divalent cations (Mg$^{2+}$, Ca$^{2+}$) and metal-chelating agents (DPA, EDTA) on the MIC of thanatin against *E. coli* XJ141026.

**Time-dependent killing**. The time-kill curves for the NDM-1 clinical isolates were identified using drop plate method according to basic microbiological protocol[31]. Bacterial cells were diluted to $1 \times 10^6$ CFU/mL and incubated with thanatin or ceftazidime for 0, 0.5, 1, 3 or 6 h. At each time point, aliquots of each culture were collected, diluted and plated onto agar plates. After overnight culture, the CFU number was calculated from the colonies growing on the plates, and an untreated inoculum group was used as the negative control.

**Sample preparation for membrane permeability assay**. Thanatin-induced outer and inner membrane permeability was measured as previously described[29]. In brief, 10 mL of LB broth was inoculated with 0.1 mL of saturated *E. coli* XJ141026 culture and grown at 37 °C overnight. Strains were diluted to $10^8$ CFU/mL, and then thanatin was added to cell cultures to 0.8 μM. An equal volume of sterile PBS was added as the control. The strains were collected, washed, and resuspended in 5 mM HEPES and 5 mM glucose buffer (pH = 7.2) at different time points.

**Outer membrane permeability**. Outer membrane permeability of thanatin was measured by the uptake of 1-N-phenylnaphthylamine (NPN)[32]. The prepared

samples were incubated with NPN (8 µL from a 500 µM stock in acetone) for 30 min at 25 °C. The samples were then transferred to cuvettes. Fluorescence was measured using an F-2500 fluorescence spectrophotometer (Hitachi, Japan) at an excitation wavelength of 350 nm and an emission wavelength of 420 nm.

**Inner membrane permeability**. The inner membrane permeability of thanatin was measured by the uptake of PI[33]. In brief, 10 µM PI was added to the cells and incubated for 30 min at 25 °C. The fluorescence of the dye was monitored using a fluorescence spectrophotometer at an excitation wavelength of 535 nm and an emission wavelength of 617 nm.

**Scanning electron microscopy**. E. coli XJ141026 culture at mid-logarithmic growth-phase was diluted in MHB to $1 \times 10^8$ CFU/mL, cultured with 0.8, 1.6 and 3.2 µM thanatin at 230 rpm for 4 h, harvested, and washed. As a control, bacteria were exposed to MHB without the peptide. Some specimens were observed in a scanning electron microscope (Hitachi S-3400N, Japan), and images were recorded.

**Detection of $Ca^{2+}$ and LPS**. The strain E. coli XJ141026 was tested in PBS without $Ca^{2+}$ and $Mg^{2+}$. Bacteria were grown to stationary phase, centrifuged, washed, resuspended in PBS ($3 \times 10^{10}$ CFU/mL), and incubated with 13 or 26 µM thanatin at 37 °C. Aliquots of thanatin-treated and untreated cultures were obtained at 0, 0.5, 1, 3, and 6 h. Then, 50 µL of the culture was used to detect CFUs. The supernatant from the remaining cell suspension was collected by centrifugation and filtered through a 0.22 µm filter (Millipore, USA). LPS release was assessed using a chromogenic limulus amebocyte lysate assay (Xiamen Bioendo Technology, Co., Ltd., China)[34]. The amount of $Ca^{2+}$ was quantified using an Amplite Colorimetric Calcium Quantitation Kit (AAT Bioquest Inc., California, USA), which has a dye that changes color when bound to $Ca^{2+}$. The analyses were conducted in 96-well microtiter plates according to the manufacturer's protocols.

**Effects of excess $Ca^{2+}$ on bacterial LPS release and CFUs**. E. coli XJ141026 was tested in PBS with 0.1, 0.5, and 1 mM $Ca^{2+}$ in the absence or presence of 13 or 26 µM thanatin to explore the effect of $Ca^{2+}$ availability on LPS release from cells treated with thanatin. Bacterial counts and LPS release were detected after 6 h as described above.

**Isothermal titration calorimetry (ITC)**. Microcalorimetric measurements of the binding of thanatin or divalent cations ($Mg^{2+}$, $Ca^{2+}$) to LPS were performed on a MicroCal Auto-ITC200 instrument (Malvern Instruments, Malvern, UK)[35]. LPS (E. coli serotype 055:B5, Sigma, USA) was dissolved in 20 mM Tris-HCl (pH = 6.8) or 10 mM PBS (pH = 7.4), vortexed vigorously for 15 min, and sonicated for 15 min at 60 °C. The LPS solution was sonicated for 5 min prior to use. Thanatin was dissolved in Tris-HCl (pH = 6.8) and titrated into LPS in Tris-HCl (pH = 6.8). Divalent cations ($Mg^{2+}$, $Ca^{2+}$) were dissolved in PBS (pH = 7.4) and titrated into LPS in PBS (pH = 7.4). All samples were degassed for 10 min in a sonication bath before the experiments. These experiments were performed at 25 °C. The generated peaks were integrated using Origin 7.0 software. The errors for all the reported thermodynamic parameters were estimated through Monte Carlo simulation with the standard errors of three experiments.

**Detection of NDM-1 protein levels and hydrolytic activity**. An overnight culture of E. coli XJ141026 was diluted to 1:2 in LB and incubated at 37 °C with 0.8, 1.6, and 3.2 µM thanatin or colistin under shaking at 250 rpm for 6 or 12 h. At given intervals, the culture supernatant was collected by centrifugation at 12,000 rpm and 4 °C for 20 min and filtered through 0.22 µm filters to eliminate bacteria. NDM-1 level in the filtered supernatant was measured using a specific ELISA kit (Shanghai Enzyme-linked Biotechnology Co., Ltd., China) and western blot analysis. Imipenem hydrolysis in the filtered supernatant at 6 h was monitored using a Biotek powerwave HT microplate spectrophotometer at 30 °C by determining the changes in absorbance at 300 nm, with a final substrate concentration of 100 µM.

Thanatin- and colistin-treated E. coli XJ141026 cell precipitates were harvested at 6 h and washed twice with 20 mM Tris-HCl (pH = 7.4). The washed cells were resuspended in the lysis buffer (Jiangsu KeyGEN BioTECH Corp., Ltd., China) supplemented with 1 mM PMSF, 1 mM DTT at 4 °C for 10 min, and disrupted by sonication. Cell debris was removed by centrifugation at 12,000 rpm and 4 °C for 20 min. Soluble total protein concentration was determined and normalized with the Pierce BCA Protein Assay Kit (Thermo Fisher Scientific, Waltham, MA, USA). NDM-1 levels in the cells were measured by SDS-PAGE followed by western blot analysis. The primary antibody against NDM-1 was obtained from Detai Bio-Tech Co., Ltd. by immunizing New Zealand white rabbits with purified NDM-1 protein and further detected by horseradish peroxidase-conjugated anti-rabbit secondary antibody (Santa Cruz Biotechnology Inc., USA). Anti-GroEL antibody (Abcam, ab82592) was used to verify protein loading transferred to the membranes. To clearly normalize the levels of NDM-1, we investigated the two proteins on the same membrane. After detecting the bands of NDM-1, the stripping buffer was used to remove the primary and secondary antibodies from the membrane according to the manufacturer's protocol (Beijing Solarbio Science & Technology Co., Ltd., China). The stripped membrane was then used to detect the levels of GroEL. Protein band intensities were quantified from polyvinylidene difluoride membranes with ImageJ software.

The cell pellet was washed and resuspended in 20 mM Tris-HCl (pH = 7.4) to evaluate imipenem hydrolysis in bacterial cell precipitates. The concentrations were equalized by adjusting the absorbance values to $OD_{600} = 0.3$ in the same buffer. After 50 µL of the diluted bacterial cell suspension was transferred to a 96-well plate, 50 µL of 200 µM imipenem was added to measure the hydrolytic activity of NDM-1 and 50 µL of 20 mM Tris-HCl (pH = 7.4) was added as blank. The assays were monitored at 300 nm by using a Biotek powerwave HT microplate spectrophotometer at 30 °C. Data were corrected by subtracting the absorbance values obtained from the blanks.

**NDM-1 inhibition assays**. $IC_{50}$ value and $K_i$ were detected to evaluate the inhibitory effect of thanatin. Thanatin was dissolved in 50 mM HEPES (pH = 6.8) containing 0.05% Tween-20 to a final concentration of 800 µM and diluted in the same buffer. The purified NDM-1 protein dissolved in 50 mM HEPES (pH = 7.4) containing 0.05% Tween-20 supplemented with 10 µM $ZnSO_4$ was pre-incubated with different concentrations of thanatin for 20 min at 30 °C. The assay was initiated by adding imipenem to the mixture of NDM-1 and thanatin, with a final enzyme concentration of 1 nM, imipenem of 100 µM and 12 different concentrations of thanatin. The change in absorbance at 300 nm was monitored for 30 min in 96-well plates at 30 °C. The initial rate of reaction for each inhibitor concentration was calculated from the slope of the initial linear phase of the respective time course. The $IC_{50}$ value was obtained by plotting the residual enzyme activity on imipenem (%) vs. inhibitor concentration ($log_{10}$). $K_i$ was obtained by nonlinear fitting of the initial rates ($V_0$) of the hydrolysis of imipenem (100, 200, 400, 800, and 1600 µM) by 1 nM NDM-1 in the absence and presence of three concentrations of thanatin. The best fits were obtained with the competitive inhibition model by using the equation $V_0 = (V_{max} \times [S]) / (K_m \times (1 + [I] / K_i) + [S])$. $K_m$, $[I]$, and $[S]$ in the equation correspond to Michaelis–Menten constant, thanatin concentration, and imipenem concentration, respectively. The kinetics parameters $K_m / V_{max}/K_{cat}$ were calculated by fitting the data into the double reciprocal Lineweaver–Burk plots. GraphPad 5.0 was used for data analysis.

**$Zn^{2+}$ restoration assays**. Purified-NDM-1 (4 nM) supplemented with 10 µM $ZnSO_4$ was mixed with thanatin (200 µM) in 1:1 volume ratio. After incubating for 20 min at 30 °C, 50 µL of the mixture was added to the 96-well plate. The total volume was adjusted to 100 µL with the following final concentrations: 1 nM NDM-1, 50 µM thanatin, 100 µM imipenem and $ZnSO_4$ ranging from 0.1 µM to 800 µM. The absorbance at 300 nm was monitored for 30 min at 30 °C. Percent residual activity was calculated from the thanatin-free control[28].

**Microscale thermophoresis (MST)**. The $K_d$ values of the binding of NDM-1 to $Zn^{2+}$, thanatin and colistin were measured using a Monolith NT.115 Pico (Nanotemper Technologies GmbH, Munich, Germany)[36,37]. Apo-NDM-1 carrying a polyhistidine-tag (His-tag) was labeled with the RED-tris-NTA second-generation dye for 30 min at room temperature in the dark according to the manufacturer's instructions (Monolith His-tag Labeling Kit RED-tris-NTA 2nd generation, Nanotemper Technologies GmbH). $Zn^{2+}$, thanatin, and colistin solutions were serially diluted in the reaction buffer (50 mM HEPES buffer [pH = 7.4] containing 0.05% Tween-20 for $Zn^{2+}$ and colistin; 50 mM HEPES [pH = 6.8] containing 0.05% Tween-20 for thanatin). Then, 100 nM labeled NDM-1 protein was added to the serial dilution of the compound in a 1:1 volume ratio. After incubation for 30 min at room temperature, the samples were examined with Monolith NT.115 in Premium Capillaries (MO-K025) at medium MST power and 5% LED/excitation power. $K_d$ values were determined using the Monolith Affinity Analysis $K_d$ fit from triplicate experiments.

**Preparation of holo-NDM-1 and ICP-MS assay**. To prepare holo-NDM-1[38], we incubated the purified NDM-1 with 2.5 molar equivalents of $Zn^{2+}$ in the ICP-MS buffer (50 mM HEPES, 200 mM NaCl, pH = 7.4) for 1 h on ice. The reconstituted holo-NDM-1 was dialyzed (10 kDa MWCO) overnight at 4 °C against 100 volumes of Chelex-treated ICP-MS buffer containing 0.05% Tween-20 and desalted in the same buffer by using 7 kDa MWCO Zeba™ Spin Desalting Columns (Thermo Fisher Scientific) to remove excess $Zn^{2+}$.

Following the exchange, the holo-NDM-1 was diluted to 4 µM and incubated with 0, 20, 40, and 80 µM thanatin for 3 h at room temperature with gentle shaking. The thanatin-treated holo-NDM-1 was buffer-exchanged again to remove any unbound $Zn^{2+}$ and thanatin as described above. The final protein concentration was determined using the NDM-1 ELISA kit (Shanghai Enzyme-linked Biotechnology Co., Ltd., China). The final protein was diluted to 1 µM and digested in 5% nitric acid at 80 °C for 1 h. The digested samples were measured with a Perkin Elmer NexION™ 350D ICP-MS[28,38]. The concentrations of $Zn^{2+}$ were calculated according to the calibration curve, and data were corrected by subtracting the values of $Zn^{2+}$ concentration obtained from the background control buffer. Finally, the molar equivalents of $Zn^{2+}$ were calculated from the ratio of $Zn^{2+}$ concentration to the enzyme concentration.

**Fractional inhibitory concentration (FIC)**. Synergy in vitro was assessed by checkerboard assays. The plates were set up with serial doubling dilutions of thanatin and meropenem or imipenem at various concentrations. Following incubation, the synergistic/additive effect was determined by calculating the FIC indices (FICI) according to the formula: FICI = (MIC of A in combination/MIC of A) + (MIC of B in combination/MIC of B). The synergy or additive was defined according to standard criteria (FICI ≤ 0.5 was defined as synergistic; 0.5 < FICI ≤ 1 was defined as additive; 1 < FICI ≤ 4 was defined as indifference; FICI > 4 was defined as antagonism)[39].

The kill curves were determined as mentioned above to investigate the bactericidal activity of the combination treatment. Bacterial cells were diluted to $1 \times 10^6$ CFU/mL and incubated with thanatin, carbapenem or their combination for 16 h. CFUs were counted.

**Cytotoxicity**. HUVECs (ATCC-CRL1730) were cultured in Dulbecco's modified Eagle medium (DMEM) containing 10% fetal bovine serum (FBS) and penicillin–streptomycin (100 units/mL). HPAEpiCs (ScienCell Research Lab., Catalog#3200, San Diego, CA) were incubated in alveolar epithelial cell medium (AEpiCM, ScienCell Research Lab., Catalog#3201). HUVECs and HPAEpiCs were seeded into 96-well plates at $5 \times 10^3$ cells/well for 24 h. The cells were washed twice with PBS and once with the corresponding serum-free culture medium. The cells were treated with a range of concentrations of thanatin or colistin in the serum-free culture medium for 24 h. Cytotoxicity was analyzed using Cell Counting Kit (CCK-8) (Dojindo, Kumamoto, Japan).

Mouse primary cortical neurons were isolated and plated at a density of $1.5 \times 10^5$ cells/cm$^2$ in wells pre-coated with poly-L-lysine (25 μg/mL) and seeded in DMEM containing 10% FBS in a humidified 5% $CO_2$ incubator at 37 °C. The DMEM was replaced with Neurobasal medium (Gibco-12348-017, Thermo Fisher Scientific, Waltham, MA, USA) supplemented with 2% B27 (Gibco-17504044) after 4 h of plating. The cells were used for experiments after culturing for 7 d in an incubator under constant temperature and humidity. The cells were treated with 50 μM thanatin for 24 h, with the addition of an equal volume of culture medium as a negative control and 0.1% Triton X-100 as a positive control. PI and Hoechst 33342 were used to detect plasma membrane permeability and cell death. The fluorescence of the dye was observed with an inverted fluorescence microscope (ECLIPSE Ti-S, Nikon, Japan).

**Animal studies**. Male BALB/c mice aged 8–10 weeks and weighing 18–22 g were used. The experimental and animal care procedures were approved by the animal care and use committee of the Fourth Military Medical University.

**Mouse pneumonia model**. The mice were anesthetized with 10 mg/mL pento-barbital via intraperitoneal injection at a dose of 60 mg/kg and intranasally infected with $1 \times 10^9$ CFU E. coli XJ141026 or K. pneumoniae XJ155017 in 30 μL of MHB. After bacterial challenge, E. coli XJ141026-infected mice were intraperitoneally injected with 6 mg/kg thanatin at 1 and 6 h. K. pneumoniae XJ155017-infected mice were intraperitoneally injected with 3, 6, or 9 mg/kg thanatin at 1 and 6 h. Bacterial numbers in the lungs were recorded at 24 h post infection as described above. The left lungs were fixed with 10% formaldehyde and embedded in paraffin for hematoxylin and eosin (H&E) staining. The survival of 10 mice in each group was monitored for 7 d after infection. Cumulative survival rate was determined.

BALF was collected after thanatin treatment at 1 and 6 h and filtered through a 0.22 μm filter. The bacterial numbers and LPS levels were measured as described above.

**Mouse sepsis model**. A sepsis model was established through an intraperitoneal administration of $4 \times 10^7$ CFU E. coli XJ141026 in 0.1 mL of MHB. After bacterial challenge, the mice were treated with a 1, 3, or 6 mg/kg thanatin at 1 and 6 h. Six mice in each group were anesthetized with 10 mg/mL pentobarbital via intraperitoneal injection at a dose of 60 mg/kg to assess bacterial clearance. Afterward, 100 μL of the blood samples were collected from retro-orbital sinus after bacterial challenge for 24 h. The lungs, livers, and spleens were harvested aseptically from each sacrificed animal. These organs were weighed and homogenized in a sterile saline solution. The homogenates were plated onto agar plates at appropriate diluted concentrations. Colonies were counted after 18 h of incubation at 37 °C. Colony counts in the tissues and blood were expressed as CFU/g. The survival of 10 mice in each group was monitored for 7 days after infection. Cumulative percentage survival was calculated. Parts of the lung, liver, and spleen were harvested, washed with sterile PBS, and fixed in 10% formaldehyde for 24 h. Their morphologies were observed using H&E staining.

**Synergism of thanatin and meropenem in the sepsis model**. BALB/c mice were infected intraperitoneally with $8 \times 10^7$ CFU of E. coli XJ141026 in 0.1 mL of MHB. The mice were treated with 10 mg/kg meropenem, 0.1 mg/kg thanatin, or their combination at 1 h post challenge. The model group was intraperitoneally injected with an equivalent volume of PBS. Six mice in each group were sacrificed after bacterial challenge for 24 h to assess bacterial clearance in the spleens and livers. In addition, the survival rates were detected as described above.

**Statistical analysis**. The results are expressed as means ± s.e.m. Unpaired two-tailed $t$-tests, one-way ANOVA, two-way ANOVA or log-rank test were used for statistical evaluations. A probability ($P$) value of 0.05 was considered indicative of statistical significance.

**Reporting summary**. Further information on research design is available in the Nature Research Reporting Summary linked to this article.

## Data availability
The source data underlying Figs. 1a–d, 2, 3a–d, 4, 5, and 6b–f, and Supplementary Figs. 1, 2, 3a–d, 4a, 7a, b, 8a–f, 9, 10, 11, and 12 are provided in the Source Data file. The original data of the unprocessed blot and gel images are also available in the Source Data file. The source data can also be found at figshare (https://doi.org/10.6084/m9.figshare.8481098.v1). All other data that support the findings of this study are available from the corresponding author upon request.

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

## Acknowledgements

We thank Xiuli Xu and Shan Zhou for providing us with NDM-1 strains from the Department of Clinical Laboratory Medicine of Xijing Hospital. This work was supported by grants from the National Natural Science Foundation of China (no. 81673477, 81471997, and 81001460).

## Author contributions

Z.H., B.M., and X.X.L. designed the experiments; Y.Z. and M.Z.W. accomplished the expression and purification of NDM-1 protein; B.M., L.S.L., M.Z.W., and X.Y.X. performed the experiments; Y.H. and M.K.L analyzed the data; C.F., Z.H., and B.M. wrote the manuscript. All authors reviewed the manuscript.

## Additional information

**Competing interests:** The authors declare no competing interests.

