## [Peer Review File · Nature Communications]

Reviewers' comments:

Reviewer #1 (Remarks to the Author):

Antibiotic resistance in pathogenic and opportunistic bacteria is continuously spreading worldwide, representing a major threat for human health. NDM-1 is a metal-dependent lactamase able to hydrolyze carbapenems, last resort antibiotics. The search for NDM inhibitors is therefore compelling.

The authors here test the the action of thanatin, a 21-residue peptide, as an antibiotic or as a co-adjuvant of carbapenems against MBL-producing bacterial strains. In this regard, the work seems like an extension of a previous article by the same authors, showing that thanatin displays antibacterial activity against ESBL-producing *E. coli* strains (ref.13). The authors surprisingly disregard a large amount of work describing the antibacterial action of thanatin, omitting even the first report in 1996 by Fehlbaum et al., PNAS.

The work would be better substantiated by providing a rationale for the use of thanatin against bacteria expressing NDM-1. Since the activity of NDM depends on the presence of two Zn(II) ions in its active site, and Ca(II) ions bind the phosphate group of the lipopolysaccharide, the authors argue that "divalent cations may be may be potential targets for the development of effective antibacterial agents to overcome the issue of severe drug resistance." (p.4). This contention is rather weak, since (1) all living organisms depend on divalent cations, and chelation therapies should be as specific as possible; (2) the chemistry of Ca(II) and Zn(II), as well as their relative abundance and availability in nature is very diverse, making it difficult to selectively target these two cations at the same time with a single chelating agent, at the risk of being non-specific; (3) it is not clear why thanatin could be a good chelating agent. The sequence of thanatin (GSKKPVPIIYCNRRGKQRM) contains many positively charged residues and is devoid of metal chelating residues, with the exception of the Cys residues, that form a disulfide bridge (Mandard et al. *Eur J Biochem.* 1998 Sep 1;256(2):404-10). Thus, there is no rationale for the use of thanatin for the selective use against NDM-expressing strains.

The authors show that thanatin can be used as an antibiotic or as a co-adjuvant of carbapenems, as shown by the impact on the MICs of NDM-expressing strains, and further supported by *in vivo* studies. However, the biochemistry that is required to provide a molecular mechanism to this action is very poor and restricts the finding to the phenomenological level. In addition, the action of thanatin against other bacterial strains has already been shown by the authors, together with its lack of toxicity. The antibacterial action of thanatin has been related to its ability to disrupt the outer membrane of bacteria and induce cell agglutination. Therefore, the purported specificity against NDM-expressing strains cannot be supported. Thus, this report is of limited value, and the molecular rationale provided are not supported by the experiments.

Experimental problems:

Figure 2. The authors show (Fig.2) that thanatin induces release of the Ca(II) ions and the LPS, thus revealing a disruption of the outer membrane. This activity by itself makes thanatin an antibacterial, such as colistin, and does not necessarily implies chelation of the Ca(II) ions, as the authors suggest in the introduction. In the next set of experiments (Fig 3), the authors conclude that thanatin "is likely to replace the divalent cations and bind with the phosphate groups on the OM." Then, the authors measure the binding affinity of thanatin to the LPS by ITC. This experiment does not demonstrate competition, but only suggests that is feasible based on the thermodynamics. The binding constants should be determined more precisely, as errors equal or exceed the determined values. Finally, is thanatin is meant to bind the Ca(II) ions as a chelating agent, it is not clear how it is able to bind strongly to the phosphate moieties in the LPS. A recent report referenced by the authors (Sinha et al. (2017), *Sci. Reports*) provides an explanation for this.

Lines 145-153. Direct evidence of binding of thanatin to the outer membrane should be assessed from membrane preparations.

Line 171. Which is the criteria for using 200 microM thanatin? Is that similar to the concentration achieved in the infected tissues of mice administered with 6 or 9 mg/kg thanatin? This should be

clarified. Also, colistin is a more potent antibiotic, so the toxicity may not necessarily be compared with the same concentration.

Lines 183-187. The authors conclude that thanatin inactivates NDM-1 by displacing the Zn(II) ions. This is not clear from the experiments herein described. Experiments from panels a-c in Figure 5 should be performed under the same conditions to allow for comparison (different concentrations and different times were employed). There is no correlation between the amount of NDM-1 released to the supernatants and the amount remaining in the precipitates. For instance, for thanatin (1.6 μ M) there is roughly a 600% increase of NDM-1 (6 h) in supernatants vs. a 30% for colistin (3.4 μ M), while the amount retained in precipitates is similar for both antibiotics. Also, it is not clear how the amount of NDM-1 released in the supernatants is measured, and why the authors use a different method when working with precipitates. Finally, a loading control should be added in Western-blot of cells.

Lines 190-194. The authors state that while NDM-1 activity in thanatin-treated groups is inhibited, this not happens with colistin. Experimental data in Figure 5 shows that the activity is inhibited in both cases. Moreover, considering that the amount of NDM-1 released in supernatants of colistin-treated cells is lower, this may imply that the inhibitory effect of colistin against NDM-1 is even higher. Hydrolysis of imipenem should be expressed as specific activity for comparison between different supernatants.

Lines 202-206. The method used for measuring dissociation constants is not reliable as it involves chemical modification of Lysine residues by NT-650 dye. Given that there are lysines in the vicinity of the metal binding site, how can the authors be sure that the binding affinities will not be altered by this modification?

Also, it is not clear if the authors use apo- or holo-NDM-1 for performing these experiments, which is critical.

NDM-1 is a membrane-anchored protein due to the presence of a lipidation sequence in its leader peptide. The authors should specify which is the variant overexpressed and purified for the biochemical experiments. Does the synthesized gene include the leader peptide?

There are also some misconceptions along the work:

Line 42. It should say "spreading of this gene" instead of "spreading of these bacteria".

Line 57. "The innate immune system responds to metallo- β -lactamases by releasing metal-chelating proteins". This assertion is not correct. The innate immune system responds to bacterial infections, and not to MBLs.

Line 65. As highlighted above, it should be clarified that targeting of Ca(II), Mg(II) and Zn(II) ions should be specific, as these metal ions are also essential for the human host.

Figure 6. The schematic depiction should show NDM-1 anchored to the outer membrane. In addition, according to the experimental data, the authors cannot assume that NDM-1 binds two molecules of thanatin.

Other issues:

Line 93 and Figure 1. Concentration units for antibiotics should be homogeneous along Figure 1 for easy comparison between different experiments.

Reviewer #2 (Remarks to the Author):

This comprehensive manuscript describes the mechanism of action of the peptide antibiotic thanatin on *E. coli* producing the NDM-1 metallo- β -lactamase. The authors are to be recognized for the extensive documentation of the microbiological and inhibitory effects of thanatin. The morphological pictures of the membrane damage caused by the peptide were quite convincing. Although the data are generally believable, the results are presented as series of raw data without appropriate interpretations in some instance. Much of the graphical data in Figures 3 and 5 would be better presented in Tables. In addition, there are some comments about the background information and the presentation of the methods and data that the authors should note. Overall,

there are some interesting observations that have been reported, but the manuscript needs greater attention to details.

Major comments

1. The title is incorrect. Data are not provided showing "...the structure of NDM-1 bacterial outer membrane". The enzyme does not have an outer membrane. Better wording: "...structure of the outer membrane in NDM-1-producing bacteria..."
2. There are many small errors in the introductory material.
 - a. Line 41. "frequent mutations" are not prevalent in NDM-1, for which there are only 19 known variants compared to hundreds of closely-related SHV or TEM or OXA beta-lactamases.
 - b. Lines 42-43. All plasmidic beta-lactamases (and other antibiotic resistance genes found on plasmids) can be transferred among species. NDM is not unique in this characteristic.
 - c. Line 47. Add "can" so that the statement reads: "mcr-1 and blaNDM-1 can coexist in E. coli..." They are not linked and do not have to coexist with each other.
 - d. Lines 53-54. "active site" not "active sites"
 - e. Line 54. Reference 8 does not refer to NDM.
 - f. Line 70. Does it really "induce" or does it "select" resistance?
 - g. Line 78. It restores activity of beta-lactam antibiotics only, not all antibiotics.
3. Line 50 and lines 249-263. The sections regarding "lack of effective drugs to treat infections with NDM-1" need to be rewritten. This summer plazomicin, eravacycline and omadacycline were all approved for use. They all have useful antibacterial activity against NDM-1-producing bacteria. In addition, in vitro data and limited in vivo data have shown that avibactam (with or without ceftazidime) combined with aztreonam provides low MICs for the beta-lactam. Thus, there are alternative therapies that have not been mentioned by the authors.
4. The figures are quite small with all the subsections of data for one section of the manuscript included in a single figure. The authors should be more selective about what is most critical for the main body of the paper and move some of the supporting data to the supplemental material.
5. The data showing kinetic data for NDM-1 in the presence of thanatin and/or Zn²⁺ are not interpreted correctly. The authors have not proven that thanatin is a competitive inhibitor, nor have they shown that thanatin is bound to NDM-1, accompanied by a loss of the zinc ions. This is a key point to the manuscript, but competitive inhibition/zinc displacement has not been established.
 - a. Reaction rates should be presented in tables as rates per micrograms of protein not in graphs. Ideally V_{max}/K_m data based on Hanes plots of the hydrolysis data or weighted fits to the Michaelis-Menten equation should be presented.
 - b. Classical kinetic plots using varying substrate and inhibitor concentrations should be used to show the mechanism by which thanatin inhibits NDM-1. There are no kinetic data provided to establish thanatin as a competitive inhibitor.
 - c. Lines 189-209. "hydrolytic efficiency" is not used appropriately. This usually refers to kinetic parameters involving K_m/V_{max}/k_{cat}. The authors apparently mean "hydrolysis rates". A more meaningful comparison of activities would take into account the amount of NDM (or total protein) that was being assayed.
 - d. Line 194. How does one measure imipenem hydrolysis in a precipitate?
6. Lines 494-493. Time dependence of the thanatin reaction: If thanatin and NDM-1 were preincubated for fixed amounts of time before the addition of imipenem, was there increased inhibition? Was there control of the timing of the additions of imipenem and Zn²⁺ in the restoration assays?
7. Line 207. The interaction of NDM-1 with colistin was only approximately 3-fold less than with thanatin. The statement that there was no interaction is not correct.
8. Figure legends need some attention.
 - a. Be sure abbreviations are defined
 - b. Change some graphs into Tables, especially in Figures 3 and 5.
 - c. Figure 4 c – what is the difference between "Control" and "Model"?
 - d. Figure 6e,f. What was the infecting organism? What was the treatment regimen?
 - e. Figure 6h. Thanatin doesn't competitively inhibit divalent cations in membranes; it displaces them.

9. Lines 293-296. Note that many metallo-beta-lactamase inhibitors work by chelating zinc ions in other enzymes in addition to beta-lactamases.

Minor editorial comments

10. Some methods are incomplete.

a. Line 357. No reference is provided for the synthesis/source for thanatin.

b. Lines 377-380. How was the purity of the NDM-1 protein determined?

c. Line 398. Provide a reference for the "drop plate" method.

d. Line 407. Provide a reference for the membrane permeability assays.

e. Line 415-425. Define NPN and PI. In each assay, does fluorescence correspond to an internal or external signal?

f. Lines 441-442. Please provide a reference for the limulus assay.

g. Line 449. What strain was used?

h. Lines 455-456. What was the source for LPS? How was it prepared?

i. Line 468. How were the cells lysed?

j. Lines 474-476. It is not clear as to what was done.

k. Line 530. Define HUVECs, DMEM.

11. Line 479. Why was this imipenem concentration used? It's about 4 times higher than the Km value from the literature. This makes it more difficult for competition to be observed kinetically.

12. Lines 504-505. Why was Tween added to the buffers? Did this affect the activity or the kinetic properties of NDM-1?

13. Line 518. It is curious that the FIC studies were done by measuring turbidity, but the MIC studies used an endpoint of lack of visible growth. Were the data consistent using each method?

14. There are several areas in the manuscript that need further attention to correct English grammar. It is recommended that the authors have a native English speaker read through the manuscript for proper usage.

Reviewer #3 (Remarks to the Author):

Bo et al. present an interesting study showing promising in vivo data for the antimicrobial activity of thanatin to fight NDM-1 producing bacteria. The authors claim that the peptide kills bacteria by Ca^{2+} and in turn LPS displacement from the outer membrane of Gram-negative bacteria in addition to blocking the activity of NDM-1. Latter represents a novel aspect particularly important for this kind of strains. However, there are some issues regarding the mechanism of action that have to be clarified and put into the context of recently published data.

Line 286: "Sinha et al. found by solution NMR that thanatin forms an antiparallel β -sheet structure with the LPS micelle, which confers increased hydrophobicity and cationicity of thanatin to LPS." In fact, in the referenced work it was shown that thanatin in complex with LPS, revealed four stranded antiparallel β -sheet in a 'head-tail' dimeric topology, and moreover that Kd of thanatin binding to LPS is 1.55 μM . This is about two orders of magnitude higher than the reported value of 0.0279 μM and would be more closely to the Kd of Ca^{2+} binding of 9.21 μM determined in this study. This poses some questions regarding the mechanism but also ITC experiments. Latter should be described and discussed in more detail also in the method section. For example, which model was used to calculate Kd, describe type of process (endo/exothermic) and compare it with Sinha et al. and other publications. This obviously depends on peptide or divalent ions. Not less important which LPS was used - strain, type - or only lipid A? Further what is the meaning of data, if error is in the range or significantly larger than the determined parameter (see Suppl. Table S3)?

Line 457: A minor question related to the preparation of LPS aggregates: Was it really sufficient just to dissolve LPS in PBS? No sonification, no vortex mixing, temperature, equilibration for hydration ...

Line 302: "In contrast to the direct binding of AMA with Zn²⁺ at the NDM-1 protease active site, thanatin replaces the Zn²⁺ from the NDM-1 protease active site." This statement is unclear and needs elaboration. If so, why should this mechanism then be specific for thanatin and not hold for other amps as well which should be capable of displacing Zn²⁺.

Line 321ff regarding model: At high concentration thanatin destroys cell membrane integrity by breaking the divalent bridge linking LPS, at sub-inhibitory concentrations thanatin reversibly binds to NDM-1, which is anchored on the inner leaflet of the OM, blocking its activity. What is the transfer rate of thanatin at sub-inhibitory concentrations to the inner leaflet and further partitioning into the cytosol to reach the active site of NDM-1? Given the strong binding of thanatin to LPS would not one expect that this is negligible?

Finally, when discussing the mode of action the authors should relate their findings to other studies on LPS/amp interaction and in particular how their mode of action for thanatin - LPS release - relates to the model of Sinha et al., who proposed a different role of thanatin namely to induce bacterial cell agglutination? Further, what about inner membrane permeability, which shows a similar time course than OM permeabilization? Is OM destruction the key event to kill Gram-negative bacteria (line 85)?

Minor:

Some parts of the Method section are insufficiently described, e.g. line 356f: "Thanatin was synthesized and purified as described above" (where?) or line 406f: "Thanatin-induced outer and inner membrane permeability was measured as previously described" (no reference given).

Ad Figures:

Many figures are crowded, at least tables should be excluded and copied into the text.

Moreover, sometimes they display a mix of results obtained from *E. coli* and *K. pneumoniae*, showing completely different data sets e.g. Fig. 2a,b,c and 2d-h. Or Fig. 3a,b would much better fit to Fig. 2a,b dealing with Ca²⁺ release from LPS caused by thanatin.

By the way, do the concentrations used for the release experiments of Ca²⁺ and LPS (Fig. 2a,b + 3a,b) correlate to an experimentally determined MIC and (2x or 0.5x) MIC for the respective high number of CFU (5x10¹⁰ according to Fig. 2c)?

Model in Fig. 6 should be displayed separately.

RESPONSE TO REVIEWERS' COMMENTS

**Reviewer #1 (Remarks to the Author):**

**Reviewer's comments:** Antibiotic resistance in pathogenic and opportunistic bacteria
is continuously spreading worldwide, representing a major threat for human health.
NDM-1 is a metal-dependent lactamase able to hydrolyze carbapenems, last resort
antibiotics. The search for NDM inhibitors is therefore compelling.

The authors here test the action of thanatin, a 21-residue peptide, as an antibiotic
or as a co-adjuvant of carbapenems against MBL-producing bacterial strains. In this
regard, the work seems like an extension of a previous article by the same authors,
showing that thanatin displays antibacterial activity against ESBL-producing *E. coli*
strains (ref.13). The authors surprisingly disregard a large amount of work describing
the antibacterial action of thanatin, omitting even the first report in 1996 by Fehlbaum
et al., PNAS.

**Answer:** Thanatin is a promising antimicrobial peptide displaying broad antibacterial
spectrum, especially against multi-drug resistant gram-negative strains. For years,
many works have been performed to explore the therapeutic potential and underlying
mechanisms of thanatin. In the previous study, we proposed that thanatin exerts
antibacterial activities against Extended-spectrum beta-lactamase (ESBL)-producing
*E. coli* strains by the mechanism of membrane permeabilization ¹. Exactly as the
reviewer said, the present work is a continuous extension of the previous article,
because we consider thanatin likely to be available when used against more
challenging drug-resistant strains such as NDM-1 producing bacteria, and the
underlying mechanisms are worth investigating.

The paper published in 1996 is a great work that researchers investigated in
detail the antimicrobial efficacy of thanatin, all-D-thanatin, C-Amidated and truncated
forms of thanatin ². This study, however, mainly focused on the mechanistic
investigation of thanatin against NDM-1 producing strains, thus we did not cite many
studies regarding antimicrobial efficacy. In the articles we previously published

regarding the analysis of antibacterial efficacy of thanatin, we quoted the 1996 report,
as well as other studies, as reference ^{1, 3, 4}. Therefore, our studies were performed
based on every piece of previous work on thanatin that we did not deliberately
disregard or omit any contribution made by other research groups. We have revised
the section of Introduction and cited the previous work reported by Fehlbaum et al. in
the revised manuscript (ref. 12) (Lines 50-51).

**Reviewer's comments:** The work would be better substantiated by providing a
rationale for the use of thanatin against bacteria expressing NDM-1. Since the activity
of NDM depends on the presence of two Zn(II) ions in its active site, and Ca(II) ions
bind the phosphate group of the lipopolysaccharide, the authors argue that “divalent
cations may be may be potential targets for the development of effective antibacterial
agents to overcome the issue of severe drug resistance.” (p.4). This contention is
rather weak, since (1) all living organisms depend on divalent cations, and chelation
therapies should be as specific as possible; (2) the chemistry of Ca(II) and Zn(II), as
well as their relative abundance and availability in nature is very diverse, making it
difficult to selectively target these two cations at the same time with a single chelating
agent, at the risk of being non-specific; (3) it is not clear why thanatin could be a good
chelating agent. The sequence of thanatin (GSKKPVPIIYCNRRTGKQRM)
contains many positively charged residues and is devoid of metal chelating residues,
with the exception of the Cys residues, that form a disulfide bridge (Mandard et al. Eur
J Biochem. 1998 Sep 1; 256(2):404-10). Thus, there is no rationale for the use of
thanatin for the selective use against NDM-expressing strains.

**Answer:** In this study, we proposed that thanatin can recover the efficacy of β -lactam
antibiotics by inactivating NDM-1 via competitively displacing zinc ions in its active
sites, which is a novel antibacterial mechanism for thanatin and has never been
reported before. We do appreciate that the reviewer thoroughly analyzed the
impossibilities of thanatin to act as a chelating agent. However, the reviewer
misunderstood the mechanisms we proposed in this study that we did not regard
thanatin as a chelating agent. We apologize for the misunderstanding. The manuscript

has been revised to make the expression clear.

**Reviewer's comments:** The authors show that thanatin can be used as an antibiotic or
as a co-adjuvant of carbapenems, as shown by the impact on the MICs of
NDM-expressing strains, and further supported by in vivo studies. However, the
biochemistry that is required to provide a molecular mechanism to this action is very
poor and restricts the finding to the phenomenological level. In addition, the action of
thanatin against other bacterial strains has already been shown by the authors,
together with its lack of toxicity. The antibacterial action of thanatin has been related
to its ability to disrupt the outer membrane of bacteria and induce cell agglutination.
Therefore, the purported specificity against NDM-expressing strains cannot be
supported. Thus, this report is of limited value, and the molecular rationale provided
are not supported by the experiments.

**Answer:** Thanks for your comments. We performed a series of mechanism
investigation to determine the mechanisms of thanatin against NDM-1 producing
strains. The results were mainly displayed in the Figures 2, 3 and 5, and thereby
concluded that thanatin can not only target to bacterial outer membrane (OM) and
induce the release of LPS, but also replace zinc ions from the active sites of NDM-1
as a competitive inhibitor.

We do apologize for causing misunderstanding. The significance of this study is
to reveal that thanatin conquers the infection caused by NDM-1-producing strains via
a novel mechanism. The antibacterial activity is not specific to NDM-expressing
strains, it is also efficient against ESBL-producing strains as we reported previously.

**Experimental problems:**

**Question 1:** Figure 2. The authors show (Fig.2) that thanatin induces release of the
Ca(II) ions and the LPS, thus revealing a disruption of the outer membrane. This
activity by itself makes thanatin an antibacterial, such as colistin, and does not
necessarily implies chelation of the Ca(II) ions, as the authors suggest in the
introduction. In the next set of experiments (Fig 3), the authors conclude that thanatin

“is likely to replace the divalent cations and bind with the phosphate groups on the
OM.” Then, the authors measure the binding affinity of thanatin to the LPS by ITC.
This experiment does not demonstrate competition, but only suggests that is feasible
based on the thermodynamics. The binding constants should be determined more
precisely, as errors equal or exceed the determined values. Finally, is thanatin is meant
to bind the Ca(II) ions as a chelating agent, it is not clear how it is able to bind
strongly to the phosphate moieties in the LPS. A recent report referenced by the
authors (Sinha et al. (2017), Sci. Reports) provides an explanation for this.

**Answer:** The OM of gram-negative bacteria is mainly consist of LPS, divalent
cations and phospholipid, among which divalent cations bridge the phosphate groups
between LPS molecules and maintain the stability and integrity of the gram-negative
bacterial OM⁵. We speculated that thanatin, as a cationic antimicrobial peptide, could
probably replace divalent cations and bind to the anionic phosphate groups. Indeed,
we observed the release of Ca²⁺ ions and LPS induced by thanatin, which indicated a
disruption of the bacterial OM. To determine whether thanatin showed higher affinity
to LPS than Ca²⁺ and Mg²⁺, we conducted isothermal titration calorimetry (ITC),
which is a commonly used method for affinity examination and is widely applied by a
large amount of studies^{6,7,8}. The ITC data showed the affinity of thanatin to LPS is
much higher than that of Ca²⁺ and Mg²⁺. In the revised manuscript, ITC experiments
have been redone and the accuracy of results were improved with decreased the error
values.

Sinha et al. observed the interaction of thanatin with LPS micelles, which reveals
the mechanism of bacterial agglutination after co-incubation⁹. As mentioned above,
the reviewer misunderstood the mechanisms we proposed in this study that we did not
regard thanatin as a chelating agent; therefore, the strong bind of thanatin to LPS is
reasonable.

**Question 2:** Lines 145-153. Direct evidence of binding of thanatin to the outer
membrane should be assessed from membrane preparations.

**Answer:** The reviewer suggested that the direct evidence of binding of thanatin to the

OM should be assessed from membrane preparations. This is a very good model to
assess the interaction between membrane and investigated molecules. However, we
consider that the data obtained directly from bacteria are more persuasive than those
from artificial conditions. We determined the release of LPS and Ca^{2+} both in vivo
and in vitro. The affinities of thanatin, Ca^{2+} , and Mg^{2+} to LPS were determined by
isothermal titration calorimetry (ITC). The equilibrium dissociation constant (K_d) of
thanatin to LPS was $1.09 \pm 0.11 \mu\text{M}$, whereas the values for Ca^{2+} and Mg^{2+} were
much higher, indicating a stronger affinity of thanatin to LPS than those of Ca^{2+} or
128 Mg^{2+} (Fig. 3e, Supplementary Fig. 5 and Table 3). All these data suggested that
thanatin competitively replaces divalent cations to bind with LPS, which disrupts the
integrity of OM and leads to bacterial death.

**Question 3:** Line 171. Which is the criteria for using 200 microM thanatin? Is that
similar to the concentration achieved in the infected tissues of mice administered with
6 or 9 mg/kg thanatin? This should be clarified. Also, colistin is a more potent
antibiotic, so the toxicity may not necessarily be compared with the same
concentration.

**Answer:** The results of in vivo studies showed that 6 to 9 mg/kg of thanatin could
effectively enhance the survival rate of both pneumonia and sepsis models. Taking
mice weighing 20 g for instance, the effective serum concentration of thanatin is
approximately 24 to 36 μM . To evaluate the toxicity of thanatin, eukaryotes
HPAEpiCs were incubated with a series of concentrations of thanatin from 6.25 to
200 μM . However, no obvious cytotoxicity of thanatin, but not colistin, was detected
even at 200 μM , which is approximately 10 times of the therapeutic concentration.
According to the results, we proposed that thanatin is a safe and efficient antibacterial
agent.

Our results showed that the antibacterial efficacy of thanatin against NDM-1
producing strains was obviously higher than that of colistin as shown in the table
below (data not shown in the submitted manuscript).

Strains	MIC (μM)	
	Thanatin	Colistin
NDM-1 E. coli XJ141015	0.8	1.7
NDM-1 E. coli XJ141026	0.8	0.85
NDM-1 E. coli XJ141047	0.8	1.7
NDM-1 K. pneumoniae XJ155019	3.2	6.8
NDM-1 K. pneumoniae XJ155020	3.2	6.8

**Question 4:** Lines 183-187. The authors conclude that thanatin inactivates NDM-1 by
 displacing the Zn (II) ions. This is not clear from the experiments herein described.

**Answer:** In the lines 161-173 of the revised manuscript, we described the
 phenomenon that the release of NDM-1 increased after incubating with thanatin and
 colistin in a concentration- and time-dependent manner (Fig. 5a). However, the
 hydrolysis of supernatants to imipenem decreased along with the increased thanatin
 treatment concentrations (Fig. 5b), indicating that thanatin probably inactivates
 NDM-1 directly, besides membrane permeabilization.

NDM-1 is a dizinc hydrolase with two Zn^{2+} ions in its active site, so we
 speculated that thanatin may inactivate NDM-1 by displacing Zn^{2+} ions. To confirm
 our hypothesis, microscale thermophoresis (MST) was used to detect the binding
 affinities of NDM-1 to Zn^{2+} , thanatin, and colistin. We measured an equilibrium
 dissociation constant (K_d) of approximately $2.04 \mu\text{M}$ for the interaction between
 NDM-1 and thanatin (Fig. 5c). A lower affinity to NDM-1 was detected with the K_d of
 $12.69 \mu\text{M}$ for Zn^{2+} and $180.43 \mu\text{M}$ for colistin, respectively (Supplementary Fig. 9a,
 b). These data indicated the thanatin performed a higher affinity to NDM-1 than Zn^{2+} .
 In addition, classical kinetic plots showed that thanatin inhibited the NDM-1 activity
 as a competitive inhibitor (Fig. 5e, Supplementary Fig. 10), and the inhibition of
 NDM-1 activity by thanatin could be reversed by increasing concentrations of Zn^{2+}
 ions (Fig. 5f). Thus, we conclude that thanatin inactivates NDM-1 by displacing the
 Zn^{2+} ions.

**Question 5:** Experiments from panels a-c in Figure 5 should be performed under the
same conditions to allow for comparison (different concentrations and different times
were employed). There is no correlation between the amount of NDM-1 released to
the supernatants and the amount remaining in the precipitates. For instance, for
thanatin (1.6 μM) there is roughly a 600% increase of NDM-1 (6 h) in supernatants vs.
a 30% for colistin (3.4 μM), while the amount retained in precipitates is similar for
both antibiotics. Also, it is not clear how the amount of NDM-1 released in the
supernatants is measured, and why the authors use a different method when working
with precipitates. Finally, a loading control should be added in Western-blot of cells.

**Answer:** Thanks for your comments. The purpose of this part of results is to show
that both thanatin and colistin damage OM integrity and cause the release of NDM-1,
rather than to compare the antibacterial efficacy between thanatin and colistin. In the
revised manuscript, the experiments in the revised manuscript have been performed
under the same condition (Lines 416-421); the concentrations of thanatin and colistin
used were 0.8, 1.6 and 3.2 μM . And the NDM-1 levels in the filtered supernatants
were measured at 6 and 12h.

The amount of NDM-1 released in the supernatants was detected by ELISA,
which is a commonly used biochemistry assay to detect the absolute value of a protein
in a liquid sample. In contrast, the amount of NDM-1 retained in precipitates was
tested by western blot. Given that western blot displays a relative value of protein
expression level, this result cannot be compared with that obtained by ELISA.

A loading control is not usually used in Western-blot of bacteria ^{10, 11}. Total
protein concentrations were tested by BCA Protein Assay, and the loading amount for
each group was determined accordingly.

**Question 6:** Lines 190-194. The authors state that while NDM-1 activity in
thanatin-treated groups is inhibited, this not happens with colistin. Experimental data
in Figure 5 shows that the activity is inhibited in both cases. Moreover, considering
that the amount of NDM-1 released in supernatants of colistin-treated cells is lower,

this may imply that the inhibitory effect of colistin against NDM-1 is even higher.
Hydrolysis of imipenem should be expressed as specific activity for comparison
between different supernatants.

**Answer:** The reviewer misinterpreted the results displayed in Figure 5. The treatment
of both thanatin and colistin caused the release of NDM-1 in a time- and
concentration-dependent manner. We predicted that the more NDM-1 is released into
the supernatant, the stronger the hydrolytic rate of the supernatant is. However, the
hydrolysis of supernatants to imipenem decreased along with the increased thanatin
treatment concentrations (Fig. 5b). These results indicated that thanatin probably
could inhibit NDM-1 directly. In contrast, treatment with colistin at high
concentrations consistently led to strong hydrolytic activity toward imipenem
(Supplementary Figure 8e), indicating that there was no obviously inhibitory effect of
colistin against NDM-1.

**Question 7:** Lines 202-206. The method used for measuring dissociation constants is
not reliable as it involves chemical modification of Lysine residues by NT-650 dye.
Given that there are lysines in the vicinity of the metal binding site, how can the
authors be sure that the binding affinities will not be altered by this modification?

**Answer:** In this study, NDM-1 protein was labeled by NT dye, and then the excess
unbound dye was eliminated following the manufacturer's instructions (Monolith
NTTM Protein Labeling Kit RED-NHS). Next, thanatin, Zn^{2+} and colistin were mixed
with the labeled NDM-1 protein, respectively. The results from different groups could
be compared since the binding affinities were detected under the same condition.

**Question 8:** Also, it is not clear if the authors use apo- or holo-NDM-1 for performing
these experiments, which is critical.

**Answer:** This is a very good question. NDM-1 protein expression was induced by
addition of 0.5 mM IPTG and 0.5 mM $ZnSO_4$ to the medium. After purified, the
holo-NDM-1 was not dialyzed against chelation buffer to obtain the apo-NDM-1. In
addition, to sustain the activity of NDM-1, the enzyme solution was supplemented

with 10 μM ZnSO_4 for NDM-1 inhibition assays and Zn^{2+} restoration assays. Thus,
we used holo-NDM-1 in our study.

**Question 9:** NDM-1 is a membrane-anchored protein due to the presence of a
lipidation sequence in its leader peptide. The authors should specify which is the
variant overexpressed and purified for the biochemical experiments. Does the
synthesized gene include the leader peptide?

**Answer:** The synthesized NDM-1 protein was used for testing its affinities to Zn^{2+} ,
thanatin and colistin, NDM-1 inhibition assays and Zn^{2+} restoration assays, rather
than to anchor the protein to the membrane. Therefore, the leader peptide was deleted
during synthesis.

**Some misconceptions and other issue figured out by the reviewer**

Line 42. It should say "spreading of this gene" instead of "spreading of these
bacteria".

Line 57. "The innate immune system responds to metallo- β -lactamases by releasing
metal-chelating proteins". This assertion is not correct. The innate immune system
responds to bacterial infections, and not to MBLs.

Line 65. As highlighted above, it should be clarified that targeting of Ca(II) , Mg(II)
and Zn(II) ions should be specific, as these metal ions are also essential for the human
host.

Figure 6. The schematic depiction should show NDM-1 anchored to the outer
membrane. In addition, according to the experimental data, the authors cannot assume
that NDM-1 binds two molecules of thanatin.

Line 93 and Figure 1. Concentration units for antibiotics should be homogeneous
along Figure 1 for easy comparison between different experiments.

**Answer:** Thanks for your kind suggestions. We appreciate that the reviewer pointed
out some typos and inaccurate expression in the manuscript, and we have modified
accordingly in the revised version.

a. Line 35. "spreading of these bacteria" was replaced by "spreading of this gene".

- b. Line 45. “The innate immune system responds to metallo- β -lactamases by releasing
metal-chelating proteins” was replaced by “The innate immune system responds to
metallo- β -lactamases producing bacteria by releasing metal-chelating proteins”.
- c. Our data showed that thanatin competitively displaced divalent cations from LPS,
and also inactivated the activity of NDM-1 enzyme by displacing zinc ions from
active sites of NDM-1. According to these results, thanatin does not act as a
chelating agent and does not target Ca^{2+} , Mg^{2+} and Zn^{2+} . Though thanatin affects
the release of divalent cations, it’s very difficult for thanatin with little cytotoxicity
to go through eukaryotic cell membranes as shown in Supplementary Fig. 7a, b of
the revised manuscript. So thanatin hardly affects the metal ions of the human host.
- 272 d. Thanks for your kind advice. We have displayed the modified schematic depiction
in Fig. 7 in the revised manuscript.
- e. In clinical test, the unit “ $\mu\text{g}/\text{mL}$ ” is usually used in MIC assay (Supplementary
Table 1, 2). In this study, however, “ μM ” was frequently used in the experiments
exploring the molecular mechanism of thanatin. So we showed the MIC of thanatin
with both “ $\mu\text{g}/\text{mL}$ ” and “ μM ” in Table 1 for easy comparison between different
experiments.

**Reviewer #2 (Remarks to the Author):**

**Reviewer’s comments:** This comprehensive manuscript describes the mechanism of
action of the peptide antibiotic thanatin on *E. coli* producing the NDM-1
metallo-beta-lactamase. The authors are to be recognized for the extensive
documentation of the microbiological and inhibitory effects of thanatin. The
morphological pictures of the membrane damage caused by the peptide were quite
convincing. Although the data are generally believable, the results are presented as
series of raw data without appropriate interpretations in some instance. Much of the
graphical data in Figures 3 and 5 would be better presented in Tables. In addition,
there are some comments about the background information and the presentation of
the methods and data that the authors should note. Overall, there are some interesting
observations that have been reported, but the manuscript needs greater attention to

details.

**Answer:** Thanks for your good advices. We have modified the presentation of our
data and provide appropriate interpretations in the revised manuscript. Meanwhile, we
will re-edit some data in Figures 3 and 5, and further improve the background
information and the presentation of the Methods.

**Major comments:**

**Question 1:** The title is incorrect. Data are not provided showing "...the structure of
NDM-1 bacterial outer membrane". The enzyme does not have an outer membrane.
Better wording: "...structure of the outer membrane in NDM-1-producing bacteria..."

**Answer:** Thanks for your kind suggestion. We have amended the title to "Cation
removal effects of thanatin on the disruption of bacterial outer membrane and
inactivation of NDM-1 enzyme".

**Question 2:** There are many small errors in the introductory material.

a. Line 41. "Frequent mutations" are not prevalent in NDM-1, for which there are
only 19 known variants compared to hundreds of closely-related SHV or TEM or
OXA beta-lactamases.

b. Lines 42-43. All plasmidic beta-lactamases (and other antibiotic resistance genes
found on plasmids) can be transferred among species. NDM is not unique in this
characteristic.

c. Line 47. Add "can" so that the statement reads: "mcr-1 and blaNDM-1 can coexist
in *E. coli*..." They are not linked and do not have to coexist with each other.

315 d. Lines 53-54. "active site" not "active sites"

e. Line 54. Reference 8 does not refer to NDM.

f. Line 70. Does it really "induce" or does it "select" resistance?

318 g. Line 78. It restores activity of beta-lactam antibiotics only, not all antibiotics.

**Answer:** We appreciate that the reviewer figured out these improper expressions in
detail. We have revised the manuscript carefully according to the reviewer's advices.

**Question 3:** Line 50 and lines 249-263. The sections regarding “lack of effective
drugs to treat infections with NDM-1” need to be rewritten. This summer plazomicin,
eravacycline and omadacycline were all approved for use. They all have useful
antibacterial activity against NDM-1-producing bacteria. In addition, in vitro data and
limited in vivo data have shown that avibactam (with or without ceftazidime)
combined with aztreonam provides low MICs for the beta-lactam. Thus, there are
alternative therapies that have not been mentioned by the authors.

**Answer:** Thanks for your kind suggestion. We have rewritten the description of the
current therapeutic situation in Discussion of the revised manuscript.

**Question 4:** The figures are quite small with all the subsections of data for one
section of the manuscript included in a single figure. The authors should be more
selective about what is most critical for the main body of the paper and move some of
the supporting data to the supplemental material.

**Answer:** Thank you for the suggestion. We have reorganized the figures to better
display our results to readers.

**Question 5:** The data showing kinetic data for NDM-1 in the presence of thanatin
and/or Zn^{2+} are not interpreted correctly. The authors have not proven that thanatin is
a competitive inhibitor, nor have they shown that thanatin is bound to NDM-1,
accompanied by a loss of the zinc ions. This is a key point to the manuscript, but
competitive inhibition/zinc displacement has not been established.

a. Reaction rates should be presented in tables as rates per micrograms of protein not
in graphs. Ideally V_{max}/K_m data based on Hanes plots of the hydrolysis data or
weighted fits to the Michaelis-Menten equation should be presented.

b. Classical kinetic plots using varying substrate and inhibitor concentrations should
be used to show the mechanism by which thanatin inhibits NDM-1. There are no
kinetic data provided to establish thanatin as a competitive inhibitor.

c. Lines 189-209. “hydrolytic efficiency” is not used appropriately. This usually refers
to kinetic parameters involving $K_m/V_{max}/k_{cat}$. The authors apparently mean

“hydrolysis rates”. A more meaningful comparison of activities would take into
account the amount of NDM (or total protein) that was being assayed.

354 d. Line 194. How does one measure imipenem hydrolysis in a precipitate?

**Answer:** Thanks very much for the constructive suggestions.

a. The kinetic parameters $K_m / V_{max} / K_{cat}$ were shown in Supplementary Table 4.

b. Classical kinetic plots were used to prove that thanatin inhibited the NDM-1
activity as a competitive inhibitor (Fig. 5e, Supplementary Fig. 10).

c. According to your kind advice, “hydrolytic efficiency” was replaced by “hydrolysis
rates” (Line 166).

361 d. We have rewritten this part in the revised Methods (Lines 433-438): “To measure
imipenem hydrolysis in cell precipitates, the cells pelleted from cultures were
washed and resuspended in 20 mM Tris-HCl (pH=7.4). The concentrations were
equalized by adjusting the absorbance values to $OD_{600}=0.3$ in the same buffer. 50
μ L diluted suspension was transferred to 96-well plates, and 50 μ L of 200 μ M
imipenem was added to measure the hydrolytic activity of NDM-1 as an observable
decrease in the absorbance of imipenem at 300 nm.”

**Question 6:** Lines 494-493. Time dependence of the thanatin reaction: If thanatin and
NDM-1 were preincubated for fixed amounts of time before the addition of imipenem,
was there increased inhibition? Was there control of the timing of the additions of
imipenem and Zn^{2+} in the restoration assays?

**Answer:** This is a very good advice. As shown in the following figure. 1 nM NDM-1
was preincubated with or without 3.125 μ M thanatin for different amounts of time.
And the assay was initiated by adding 100 μ M imipenem to the mixture. The initial
rate of hydrolysis of imipenem was evaluated. Percent residual activity was calculated
from no thanatin control. The results showed that thanatin inhibited the NDM-1
activity in a time-dependent manner before 20 min, whereas there was no significant
difference after then. Therefore, in the revised manuscript, purified NDM-1 protein
was preincubated with thanatin for 20 min in both inhibition assays and Zn^{2+}
restoration assays.

**Question 7:** Line 207. The interaction of NDM-1 with colistin was only
approximately 3-fold less than with thanatin. The statement that there was no
interaction is not correct.

**Answer:** Thank you for the suggestion. We optimized the reaction systems for MST
experiment to improve the accuracy of the results, which indicated that the affinity of
thanatin to NDM-1 is approximately 90-fold higher than that of colistin (Fig. 5c,
Supplementary Fig. 9b). We have rewritten the results in the revised manuscript
(Lines 176-181).

**Question 8:**

Figure legends need some attention.

a. Be sure abbreviations are defined.

b. Change some graphs into Tables, especially in Figures 3 and 5.

c. Figure 4 c – what is the difference between “Control” and “Model”?

398 d. Figure 6e, f. What was the infecting organism? What was the treatment regimen?

e. Figure 6h. Thanatin doesn't competitively inhibit divalent cations in membranes; it
displaces them.

**Answer:** Thanks for your kind advices.

a. We have defined abbreviations used in figures.

b. We have re-edited some data in Figures 3 and 5

c. In Figure 4, control group means normal animals with neither infection nor
treatment; whereas model group means animals infected by *E. coli* XJ141026 but
receiving no treatment. We have replaced the “control” with “normal”
(Supplementary Fig. 6).

408 d. The mice were infected with NDM-1-producing *E. coli* XJ141026, and treated
with 10 mg/kg meropenem, 0.1 mg/kg thanatin, or a combined treatment 1 h post
challenge. These above informations have been added in the revised manuscript.

(Lines 561-567)

e. We have corrected the improper expression in the legend of Fig. 7 in the revised
manuscript.

**Question 9:** Lines 293-296. Note that many metallo-beta-lactamase inhibitors work
by chelating zinc ions in other enzymes in addition to beta-lactamases.

**Answer:** We have modified the statements appropriately (Lines 254-256): “NDM-1
inhibitors, antagonizing multiple subtypes of metallo- β -lactamases and protecting
β -lactam antibiotics from being hydrolyzed, have been widely studied for synergistic
application with β -lactam antibiotics to restore their bactericidal effects.”

**Minor editorial comments**

**Question 10:** Some methods are incomplete.

a. Line 357. No reference is provided for the synthesis/source for thanatin.

b. Lines 377-380. How was the purity of the NDM-1 protein determined?

c. Line 398. Provide a reference for the “drop plate” method.

426 d. Line 407. Provide a reference for the membrane permeability assays.

e. Line 415-425. Define NPN and PI. In each assay, does fluorescence correspond to
an internal or external signal?

f. Lines 441-442. Please provide a reference for the limulus assay.

430 g. Line 449. What strain was used?

431 h. Lines 455-456. What was the source for LPS? How was it prepared?

- i. Line 468. How were the cells lysed?
- j. Lines 474-476. It is not clear as to what was done.
- k. Line 530. Define HUVECs, DMEM.
- **Answer:** Thanks for your kind advices.
- a, c, d, f. References have been provided.
- b. The purity of the NDM-1 proteins is >90%, and the result of SDS-PAGE is shown
- below.

Quality Assurance:

- e. NPN and PI are the abbreviations of 1-N-phenyl-naphthylamine and propidium
- iodide. We have defined them in the revised manuscript. NPN only becomes
- fluorescent after binding to hydrophobic regions of cell membranes¹². The tight
- packing of the fatty acyl chains of lipid A in the outer membrane leaflet limits the
- free diffusion of hydrophobic solutes, such as NPN¹³. However, once permeated,
- intercalation of NPN into the underlying phospholipid inner leaflet and the
- cytoplasmic membranes produces a resultant increase in fluorescence. PI only
- enters cells with damaged membrane and fluoresces upon binding to nucleic acids
- 448¹⁴. So NPN and PI were often used to detect the outer and inner permeability,
- respectively.
- 450 g. The strain *E. coli* XJ141026 was used in the assays of effects of excess Ca²⁺ on

bacterial LPS release and CFUs.

452 h. The detailed information of LPS (*E. coli* serotype O55:B5, Sigma, US) has been
added in the revised manuscript (Line 402).

i. The thanatin- and colistin-treated *E. coli* XJ141026 cell precipitates were harvested
at 6h, washed twice with 20 mM Tris-HCL (pH=7.4). The washed cells
resuspended in the lysis buffer (Jiangsu KeyGEN BioTECH Corp., Ltd., China)
supplemented with 1 mM PMSF, 1mM DTT at 4 °C for 10min, and disrupted by
sonication, Cell debris was removed by centrifugation at 12,000 rpm and 4 °C for
20 min (Lines 425-430).

j. The methods of detection of NDM-1 protein levels and hydrolytic activity in
NDM-1-producing bacteria have been revised carefully (Lines 416-438).

k. HUVECs and DMEM are the abbreviations of Human Umbilical Vein Endothelial
Cells and Dulbecco's Modified Eagle Medium. We have defined them in the
revised manuscript.

**Question 11:** Line 479. Why was this imipenem concentration used? It's about 4
467 times higher than the Km value from the literature. This makes it more difficult for
competition to be observed kinetically.

**Answer:** The concentration we used is according to a previous report¹⁰. However,
your suggestion really inspired us a lot. Thus, we adjusted the substrate concentration
from 400 μM to 100 μM in the revised manuscript, which is close to the reported Km
for imipenem (94 μM)¹⁵, and inhibitory effect of thanatin was more obvious (Fig.
5d).

**Question 12:** Lines 504-505. Why was Tween added to the buffers? Did this affect
the activity or the kinetic properties of NDM-1?

**Answer:** Tween-20 is a kind of detergent, which is widely used in MST assay at 0.05%
concentration to stabilize proteins and avoid aggregation formation in the solution¹⁶.
479^{17, 18}. In the revised manuscript, we tested the hydrolytic activity of NDM-1 to

480 imipenem in 50 mM HEPES (pH=7.4) containing 0.05% Tween-20. The K_m value is
481 103.80 ± 10.34 (Supplementary Table 4.), which is consistent with the reported K_m
for imipenem ($94 \mu\text{M}$)¹⁵. Thus, it does not affect the activity and the kinetic
properties of NDM-1.

**Question 13:** Line 518. It is curious that the FIC studies were done by measuring
turbidity, but the MIC studies used an endpoint of lack of visible growth. Were the
data consistent using each method?

**Answer:** In this study, FIC assay was performed to assess the synergy between drugs
in vitro, whereas MIC assay examined the antibacterial efficacy of drugs. The purpose
of both experiments is different, but they all rely on the detection of turbidity to
determine the bacterial growth status. Actually, “lack of visible growth” means the
turbidity is invisible and clear. We have unified the expression in the revised
manuscript (Lines 334, 489).

**Question 14:** There are several areas in the manuscript that need further attention to
correct English grammar. It is recommended that the authors have a native English
speaker read through the manuscript for proper use.

**Answer:** The manuscript has been revised by American Journal Experts Company
(<http://www.aje.com/>).

**Reviewer #3 (Remarks to the Author):**

**Reviewer’s comments:** Bo et al. present an interesting study showing promising in
vivo data for the antimicrobial activity of thanatin to fight NDM-1 producing bacteria.
The authors claim that the peptide kills bacteria by Ca^{2+} and in turn LPS displacement
from the outer membrane of Gram-negative bacteria in addition to blocking the
activity of NDM-1. Latter represents a novel aspect particularly important for this
kind of strains. However, there are some issues regarding the mechanism of action
that have to be clarified and put into the context of recently published data.

**Answer:** Thanks very much for your positive comments on our in vivo data regarding
the antimicrobial activity of thanatin to fight NDM-1 producing bacteria. We are also
very glad to have this precious chance to in-depth analyze our data with experts in this
field.

**Question 1:** Line 286: “Sinha et al. found by solution NMR that thanatin forms an
antiparallel β -sheet structure with the LPS micelle, which confers increased
hydrophobicity and cationicity of thanatin to LPS.” In fact, in the referenced work it
was shown that thanatin in complex with LPS, revealed four stranded antiparallel
β -sheet in a ‘head-tail’ dimeric topology, and moreover that K_d of thanatin binding to
LPS is 1.55 μM . This is about two orders of magnitude higher than the reported value
of 0.0279 μM and would be more closely to the K_d of Ca^{2+} binding of 9.21 μM
determined in this study. This poses some questions regarding the mechanism but also
ITC experiments. Latter should be described and discussed in more detail also in the
method section. For example, which model was used to calculate K_d , describe type of
process (endo/exothermic) and compare it with Sinha et al. and other publications.
This obviously depends on peptide or divalent ions. Not less important which LPS
was used - strain, type - or only lipid A? Further what is the meaning of data, if error
is in the range or significantly larger than the determined parameter (see Suppl. Table
S3)?

**Answer:** We apologize for the misunderstanding caused by our unclear expression. In
the previous manuscript, the K value in the ITC experiments was the K_a value
(association constant) that higher K_a value represents stronger affinity between two
molecules. Contrarily, the K_d value in Sinha’s paper is dissociation constant that lower
K_d means stronger affinity. The association constant is the inverse of the dissociation
constant, so thanatin’s affinity K_d is about 3.58 μM ($1/2.79 \times 10^5 \text{ M}^{-1}$). In the revised
manuscript, we modified the reaction system of ITC, the equilibrium dissociation
constant (K_d) of thanatin to LPS was $1.09 \pm 0.11 \mu\text{M}$, which is similar with Sinha’s in
vitro data (1.55 μM). The K_d values of Ca^{2+} / Mg^{2+} are much higher than that of

thanatin. We have marked the K value more clearly in the revised manuscript and
more detailed information has been added in the method and discussion sections in
the revised manuscript.

Thanks very much for your carefulness. The detailed information of LPS (*E. coli*
serotype O55:B5, Sigma, US) has been added in the revised manuscript (Line 402).

We have optimized the reaction systems for ITC experiment to improve the
accuracy of the results with decreased the error values (Supplementary Table 3).

**Question 2:** Line 457: A minor question related to the preparation of LPS aggregates:
Was it really sufficient just to dissolve LPS in PBS? No sonification, no vortex mixing,
temperature, equilibration for hydration ...

**Answer:** Thanks very much for your carefulness. The description of LPS preparation
have been added in the Methods section in the revised manuscript: LPS (*E. coli*
serotype 055:B5, Sigma, US) was dissolved in 20 mM Tris-HCl (pH=6.8) or 10 mM
PBS (pH=7.4), vortexed vigorously for 15 min, and then sonicated for 15 min at
60 °C. Samples were then sonicated for 5 min prior to use (Lines 402-405).

**Question 3:** Line 302: “In contrast to the direct binding of AMA with Zn^{2+} at the
NDM-1 protease active site, thanatin replaces the Zn^{2+} from the NDM-1 protease
active site.” This statement is unclear and needs elaboration. If so, why should this
mechanism then be specific for thanatin and not hold for other amps as well which
should be capable of displacing Zn^{2+} .

**Answer:** Thanks very much for your constructive advice. More detailed explanation
and discussion have been added in our revised manuscript (Lines 258-264).

This is a very good question, which has aroused our attention before. We have
thought it over for a long time, but appropriate methods to investigate this mechanism
have not been found. Sinha’s data, which revealed the structure of thanatin in complex
with LPS, gave us some inspiration. Thanatin forms four stranded antiparallel β -sheet
in a ‘head-tail’ dimeric topology, and displays higher hydrophobicity and cationicity
with sites of LPS interactions⁹. Therefore, we speculated that the formation of special

dimeric structure of thanatin may be the primary cause of zinc ions replacement and
NDM-1 inactivation. Thanatin in free solution assumed an antiparallel β -hairpin
conformation, which promotes the formation of dimeric structure⁹. While colistin is a
cyclopeptide, which may be hard to close and arrive the active sites of NDM-1 protein.
More in-depth researches need to be done.

**Question 4:** Line 321ff regarding model: At high concentration thanatin destroys cell
membrane integrity by breaking the divalent bridge linking LPS, at sub-inhibitory
concentrations thanatin reversibly binds to NDM-1, which is anchored on the inner
leaflet of the OM, blocking its activity. What is the transfer rate of thanatin at
sub-inhibitory concentrations to the inner leaflet and further partitioning into the
cytosol to reach the active site of NDM-1? Given the strong binding of thanatin to
LPS would not one expect that this is negligible? Finally, when discussing the mode
of action the authors should relate their findings to other studies on LPS/amp
interaction and in particular how their mode of action for thanatin - LPS release -
relates to the model of Sinha et al., who proposed a different role of thanatin namely
to induce bacterial cell agglutination?

**Answer:** Lisandro *et al* have confirmed that NDM-1 is a lapidated protein that
anchors to the outer membrane of Gram-negative bacteria, which contributes to the
unusual stability of NDM-1¹⁰. Given the distribution of NDM-1, anchoring to the
outer membrane, we speculated that thanatin probably could easily reach the active
site of NDM-1. Our new ITC and MST results showed that thanatin had a stronger
binding capacity to LPS ($K_d = 1.09 \pm 0.11 \mu\text{M}$) (Fig. 3e) and NDM-1 ($K_d = 2.04 \pm 0.55$
μM) (Fig. 5c), the happening of actual binding depends on many intricate factors,
such as the accessibility of NDM-1 active sites, the bridging strength between LPS
and divalent cations, and the fluidity of bacterial outer membrane, et al. Based on our
data, thanatin killed most NDM-1 producing *E. coli* (5×10^5 CFU/mL) via membrane
permeabilization at MIC. However, at concentrations much lower than the MIC
values obtained by monoadministration, thanatin greatly enhanced the bactericidal
capacities of meropenem and imipenem against the NDM-1-producing *E. coli*

XJ141026 (Fig. 6b, c). At subinhibitory concentrations, thanatin could hardly affect
the growth of NDM-1-producing strains. Therefore, we considered that thanatin
mainly inactivated NDM-1, rather than exerting antibacterial activity, when used at
low concentrations.

Thanks very much for your constructive advice. We have observed the
agglutination of *E. coli* ATCC35218 after thanatin treatment in 2011¹. Sinha wrote in
the Introduction section that agglutinated cells by thanatin treatment could be
efficiently removed by phagocytosis without releasing toxic substances into systemic
circulation⁹. Moreover, Sinha cited three papers to confirm that some AMPs exert
antibacterial activity by inducing bacterial cell agglutination^{19, 20, 21}. However, these
three papers did not mention how agglutination causes bacterial death. If thanatin
could only exert agglutination action, it would be hard to explain its bactericidal
ability, which has been widely proved by our and others' studies^{1, 2, 3, 4}. Sinha's
studies well confirmed the interaction between thanatin and LPS in vitro. Our studies
firstly confirmed in vivo that thanatin competitively displaces divalent cations,
induces LPS release, and thereafter disrupts the outer membrane of NDM-1 producing
*E. coli*. Based on current studies, we speculated that bacterial agglutination may be an
earlier phenomenon occurring before membrane permeabilization, and they are not
contradicted. We very appreciate your positive comments that we present an
interesting study showing promising in vivo data for the antimicrobial activity of
thanatin to fight NDM-1 producing bacteria.

**Question 5:** Further, what about inner membrane permeability, which shows a similar
time course than OM permeabilization? Is OM destruction the key event to kill
Gram-negative bacteria (line 85)?

**Answer:** Our data showed that inner membrane permeability also increased after
incubating with thanatin in a time-dependent manner (Fig. 1d). The intracellular
fluorescence intensity of NPN and PI increased 1 and 2 h post incubation, respectively
(Fig. 1c, d). These data indicate that the increasing destruction of OM happens earlier
than the change of inner membrane permeability. Given the destruction of outer and

inner membrane occurs sequentially, we consider OM destruction as the first and
indispensable step to kill Gram-negative bacteria.

**Question 6:** Minor: Some parts of the Method section are insufficiently described, e.g.
line 356f: “Thanatin was synthesized and purified as described above” (where?) or
line 406f: “Thanatin-induced outer and inner membrane permeability was measured
as previously described” (no reference given). Ad Figures: Many figures are crowded,
at least tables should be excluded and copied into the text. Moreover, sometimes they
display a mix of results obtained from *E. coli* and *K. pneumoniae*, showing
completely different data sets e.g. Fig. 2 a, b, c and 2d-h. Or Fig. 3a, b would much
better fit to Fig. 2a, b dealing with Ca^{2+} release from LPS caused by thanatin.

**Answer:** We appreciate the reviewer’s kind suggestion. We apologize that some
detailed description was not supported by references. We have cited related studies
and reorganized the figures to better display our results to readers in the revised
manuscript.

To verify that thanatin promotes the release of Ca^{2+} and LPS from different
bacterial outer membrane, *E. coli* and *K. pneumoniae* were used in our study in vitro
and in vivo, respectively. However, we neglected the consistency of the experimental
design. The LPS levels and bacterial loads in the BALF of *E. coli*-infected mice have
been tested, and the results have been presented in Fig.2 d-f and Supplementary Fig. 4
in the revised manuscript.

Although experiments are similar in Fig. 2a, b and Fig. 3a, b, the scientific issues
revealed are different. Data in Fig. 2a, b showed that thanatin destroys bacterial outer
membrane by promoting the release of Ca^{2+} and LPS. Data in Fig. 3a, b showed that
thanatin-induced LPS release was inhibited by Ca^{2+} , which indicated that thanatin
may exert a competitive antagonistic effect against Ca^{2+} . Therefore, we consider it
would be better that these two figures are separated.

**Question 7:** By the way, do the concentrations used for the release experiments of
Ca^{2+} and LPS (Fig. 2a, b + 3a, b) correlate to an experimentally determined MIC and

(2x or 0.5x) MIC for the respective high number of CFU (5×10^{10} according to Fig.
2c)?

**Answer:** This is a very good question. For the release experiments of Ca^{2+} and LPS,
if low *E. coli* CFU was used, most *E. coli* would be killed by thanatin quickly, leading
to a very small quantity of Ca^{2+} release, which was undetectable. Thus, the bacterial
load used in the MIC assay was 5×10^5 CFU/mL according to the broth microdilution
guideline of Clinical and Laboratory Standards Institute²², whereas those used in the
release experiments of Ca^{2+} and LPS were above 10^{10} CFU/mL. The MIC of thanatin
against NDM-1 *E. coli* XJ141026 is 0.8 μM ; in contrast, the concentrations of
thanatin used for the release experiments of Ca^{2+} and LPS were 13 and 26 μM ,
respectively. These concentrations were much higher than the MIC value due to the
extremely high bacterial load used in the experiments. The bacterial CFUs and the
concentrations of thanatin were optimized to accurately detect the release of Ca^{2+} and
LPS. As a result, 3×10^{10} CFU/mL *E. coli* was used in Fig.2a-c and 1×10^{10} CFU/mL *E.*
*coli* was used in Fig.3a-b. Under this condition, only about 11.95 μM Ca^{2+} was
detected in 26 μM thanatin treatment group 6 h post incubation. According to the
instruction of Amplite Colorimetric Calcium Quantitation Kit (AAT Bioquest Inc.,
California, USA), the calcium detection linear range is from 2.5 to 150 μM . Thus, we
used high number of CFU to make sure the Ca^{2+} was detectable.

**Question 8:** Model in Fig. 6 should be displayed separately.

**Answer:** Thanks for your kind advice. We have re-edited Fig. 6 and displayed the
schematic depiction in Fig. 7 in the revised manuscript.

**References**

- 1. Hou Z, Lu J, Fang C, Zhou Y, Bai H, Zhang X, *et al.* Underlying mechanism
of in vivo and in vitro activity of C-terminal-amidated thanatin against clinical
isolates of extended-spectrum beta-lactamase-producing *Escherichia coli*. *J*
*Infect Dis* 2011, **203**(2): 273-282.

- 2. Fehlbau P, Bulet P, Chernysh S, Briand JP, Roussel JP, Letellier L, *et al.*
Structure-activity analysis of thanatin, a 21-residue inducible insect defense
peptide with sequence homology to frog skin antimicrobial peptides. *Proc Natl*
*Acad Sci U S A* 1996, **93**(3): 1221-1225.
- 3. Ma B, Niu C, Zhou Y, Xue X, Meng J, Luo X, *et al.* The Disulfide Bond of the
Peptide Thanatin Is Dispensable for Its Antimicrobial Activity In Vivo and In
Vitro. *Antimicrob Agents Chemother* 2016, **60**(7): 4283-4289.
- 4. Hou Z, Da F, Liu B, Xue X, Xu X, Zhou Y, *et al.* R-thanatin inhibits growth
and biofilm formation of methicillin-resistant *Staphylococcus epidermidis* in
vivo and in vitro. *Antimicrob Agents Chemother* 2013, **57**(10): 5045-5052.
- 5. Clifton LA, Skoda MW, Le Brun AP, Ciesielski F, Kuzmenko I, Holt SA, *et al.*
Effect of divalent cation removal on the structure of gram-negative bacterial
outer membrane models. *Langmuir* 2015, **31**(1): 404-412.
- 6. Khan AU, Ali A, Danishuddin, Srivastava G, Sharma A. Potential inhibitors
designed against NDM-1 type metallo-beta-lactamases: an attempt to enhance
efficacies of antibiotics against multi-drug-resistant bacteria. *Sci Rep* 2017,
**7**(1): 9207.
- 7. Yan C, Liu D, Li L, Wempe MF, Guin S, Khanna M, *et al.* Discovery and
characterization of small molecules that target the GTPase Ral. *Nature* 2014,
**515**(7527): 443-447.
- 8. Welsch ME, Kaplan A, Chambers JM, Stokes ME, Bos PH, Zask A, *et al.*
Multivalent Small-Molecule Pan-RAS Inhibitors. *Cell* 2017, **168**(5): 878-889
e829.
- 9. Sinha S, Zheng L, Mu Y, Ng WJ, Bhattacharjya S. Structure and Interactions
of A Host Defense Antimicrobial Peptide Thanatin in Lipopolysaccharide
Micelles Reveal Mechanism of Bacterial Cell Agglutination. *Sci Rep* 2017,
**7**(1): 17795.
- 10. Gonzalez LJ, Bahr G, Nakashige TG, Nolan EM, Bonomo RA, Vila AJ.
Membrane anchoring stabilizes and favors secretion of New Delhi
metallo-beta-lactamase. *Nat Chem Biol* 2016, **12**(7): 516-522.
- 11. Gonzalez LJ, Moreno DM, Bonomo RA, Vila AJ. Host-specific
enzyme-substrate interactions in SPM-1 metallo-beta-lactamase are modulated
by second sphere residues. *PLoS pathogens* 2014, **10**(1): e1003817.
- 12. Campos MA, Morey P, Bengoechea JA. Quinolones sensitize gram-negative

- bacteria to antimicrobial peptides. *Antimicrob Agents Chemother* 2006, **50**(7):
2361-2367.
- 13. Nikaido H. Molecular basis of bacterial outer membrane permeability revisited.
*Microbiology and molecular biology reviews : MMBR* 2003, **67**(4): 593-656.
- 14. Yarlagadda V, Akkapeddi P, Manjunath GB, Haldar J. Membrane active
vancomycin analogues: a strategy to combat bacterial resistance. *Journal of*
*medicinal chemistry* 2014, **57**(11): 4558-4568.
- 15. Yuan Q, He L, Ke H. A potential substrate binding conformation of
beta-lactams and insight into the broad spectrum of NDM-1 activity.
*Antimicrob Agents Chemother* 2012, **56**(10): 5157-5163.
- 16. de Sousa LR, Wu H, Nebo L, Fernandes JB, da Silva MF, Kiefer W, *et al.*
Flavonoids as noncompetitive inhibitors of Dengue virus NS2B-NS3 protease:
inhibition kinetics and docking studies. *Bioorg Med Chem* 2015, **23**(3):
466-470.
- 17. Seidel SA, Dijkman PM, Lea WA, van den Bogaart G, Jerabek-Willemsen M,
Lazic A, *et al.* Microscale thermophoresis quantifies biomolecular interactions
under previously challenging conditions. *Methods* 2013, **59**(3): 301-315.
- 18. Jerabek-Willemsen M, Wienken CJ, Braun D, Baaske P, Duhr S. Molecular
interaction studies using microscale thermophoresis. *Assay Drug Dev Technol*
2011, **9**(4): 342-353.
- 19. Shai Y. Mode of action of membrane active antimicrobial peptides.
*Biopolymers* 2002, **66**(4): 236-248.
- 20. Choi H, Chakraborty S, Liu R, Gellman SH, Weisshaar JC. Single-Cell,
Time-Resolved Antimicrobial Effects of a Highly Cationic, Random Nylon-3
Copolymer on Live Escherichia coli. *ACS Chem Biol* 2016, **11**(1): 113-120.
- 21. Pulido D, Moussaoui M, Andreu D, Nogues MV, Torrent M, Boix E.
Antimicrobial action and cell agglutination by the eosinophil cationic protein
are modulated by the cell wall lipopolysaccharide structure. *Antimicrob Agents*
*Chemother* 2012, **56**(5): 2378-2385.
- 22. CLSI. Methods for Dilution Antimicrobial Susceptibility Tests for Bacteria
That Grow Aerobically. *Clinical Laboratory Standards Institute* 2012:
M07-A09.

Reviewers' comments:

Reviewer #1 (Remarks to the Author):

In the revised version of the present manuscript by Ma and coworkers, the authors have addressed several of the issues raised by this reviewer. The present version includes some new experiments and clarifies some aspects that were not obvious in the previous one. However, there are some important experiments and controls that the authors do not regard as important as this reviewer does.

The authors now clearly state that they attribute the inhibitory effect of thanatin against NDM-1 as resulting from displacement of the Zn(II) ions, and not by a chelating effect. Despite the efforts from the authors, the results from the studies *in vitro* do not provide information supporting this assertion, and I feel that some essential aspects of the biochemistry supporting their proposal is still largely based on speculation rather than on their experiments.

1. There are not direct evidences of metal ion displacement from NDM-1 (Question 4). The authors only show that thanatin inhibits competitively NDM-1, and then they compare several binding affinities (see below). A experiment directly measuring Zn(II) release would be required.

2. An issue of major concern to this reviewer is that the authors have determined the binding affinities of NDM-1 towards Zn(II) by using the holo-protein, i.e., already loaded with Zn(II), as they clearly state in the answers to the reviewers (lines 226-233). This means that the authors are measuring the binding of a low affinity site. In addition, they do not measure the metal content of the protein preparation.

3. Metallo-beta-lactamases requires two metal ions to be active (Gonzalez et al. (2012), *Nature Chemical Biology*, 8, 698). It is also likely that thanatin can displace only one Zn(II) ion, inactivating the enzyme. This could be a plausible explanation provided the authors measure the metal content of the enzyme and measuring the affinities using the apo-protein.

4. Lines 216-224 (Question 7). The authors clarify that they have used the same chemically modified enzyme for all the binding experiments, allowing them to compare the data. This is correct, but it does not address the issues I raised, since these are artificial conditions and the relative affinities cannot be extrapolated to the unmodified enzymes.

5. Many of the questions raised by this reviewer were answered by the authors, but they did not incorporate these aspects in the manuscript text. For example, indicating whether the *in vitro* experiments were performed with the holo or apo-protein (Question 8) or with the processed soluble enzyme (Question 9) are important for the reader. The expression "The NDM-1 gene" used in the Experimental Section (line 306) is vague, and requires detail of the N and C-terminal residues of the expressed proteins.

6. Regarding the aspect of protein quantitation in Question 5, the authors do not address our point regarding the use of different techniques. This problem could have been sorted out by performing Western blots in supernatants and precipitates. We also disagree with their assertion that loading controls are not used in Western blots. For instance, in reference 10 (cited in line 194), GroEL is used as a loading control in Figure 5.

Reviewer #2 (Remarks to the Author):

The authors have tried very hard to address the comments from the reviewers. They have revised incorrect wording as requested. The new data regarding kinetic parameters support the assertion that thanatin inhibits NDM-1 competitively.

The only remaining question deals with the measurement of imipenem hydrolysis measurements in precipitates. If a suspension of cells is used to measure hydrolysis rates, part of the decrease in absorbance must be due to the settling out of the cells during the reaction. Did the authors run appropriate blanks without imipenem so that a correction factor could be used in the calculations?

The authors should check the spelling in the manuscript and supplement. There is at least one misspelling of "imipenem" as "imipeniem".

Reviewer #3 (Remarks to the Author):

[No further comments for author.]

RESPONSE TO REVIEWERS' COMMENTS

**Reviewer #1 (Remarks to the Author):**

**Reviewer's comments:** In the revised version of the present manuscript by Ma and
coworkers, the authors have addressed several of the issues raised by this reviewer.
The present version includes some new experiments and clarifies some aspects that
were not obvious in the previous one. However, there are some important experiments
and controls that the authors do not regard as important as this reviewer does.

The authors now clearly state that they attribute the inhibitory effect of thanatin
against NDM-1 as resulting from displacement of the Zn(II) ions, and not by a
chelating effect. Despite the efforts from the authors, the results from the studies in
vitro do not provide information supporting this assertion, and I feel that some
essential aspects of the biochemistry supporting their proposal is still largely based on
speculation rather than on their experiments.

**Answer:** Thank you for your comments. We appreciate that the reviewer provided
many constructive suggestions to help us improve our study. Some important
experiments and controls were added in the revised manuscript. In brief, we have
made the following changes according to the reviewer's advice.

- 1. Apo-NDM-1 was prepared by dialysis against EDTA by using method described by
González et al.¹ and was used in microscale thermophoresis (MST) assay to detect
the binding affinities of apo-NDM-1 to Zn²⁺, thanatin, and colistin.
- 2. In the MST assay, we used RED-tris-NTA second-generation dye to label
apo-NDM-1 with polyhistidine-tag (His-tag), which could avoid the modification
of the metal binding site and improve the accuracy of the binding affinities
detected.
- 3. Holo-NDM-1 was prepared, and its metal content was measured by inductively
coupled plasma–mass spectrometry (ICP-MS) after incubation with or without
thanatin. These results directly demonstrate that thanatin causes the release of Zn²⁺
from holo-NDM-1.

- 4. Western blot analysis was performed again using GroEL as a loading control.
- 5. We made the following modifications in the revised manuscript.
- (1) We rewrote the method of the expression and purification of NDM-1. The deletion
of the signal peptide was clarified, and the protein sequence of NDM-1 was presented.
- (2) The methods of preparation of apo- and holo-NDM-1 and ICP-MS assay were
added.
- (3) The detailed information of the antibodies against GroEL and NDM-1 was added
in the method of Western blot analysis.
- (4) The method of MST assay was modified due to the new dye application.
- (5) The relevant results were added.

**Question 1:** There are not direct evidences of metal ion displacement from NDM-1
(Question 4). The authors only show that thanatin inhibits competitively NDM-1, and
then they compare several binding affinities (see below). An experiment directly
measuring Zn (II) release would be required.

**Answer:** Thank you for your kind suggestion to help us improve our study. To
directly determine whether thanatin causes the release of Zn²⁺ from NDM-1, we used
ICP-MS to measure the metal content of holo-NDM-1 after incubation with or
without thanatin. The results showed that 2.23 ± 0.28 molar equivalents of zinc are
bound to holo-NDM-1, and only 1.03 ± 0.25 molar equivalents of zinc remain bound
to NDM-1 after treatment with 20 molar equivalents of thanatin (Fig. 5f). Hence,
thanatin displaces approximately half of the zinc content from holo-NDM-1.
Therefore, we confirm that thanatin causes the release of Zn²⁺ from NDM-1.

**Question 2:** An issue of major concern to this reviewer is that the authors have
determined the binding affinities of NDM-1 towards Zn (II) by using the holo-protein,
i.e., already loaded with Zn (II), as they clearly state in the answers to the reviewers
(lines 226-233). This means that the authors are measuring the binding of a low
affinity site. In addition, they do not measure the metal content of the protein
preparation.

**Answer:** According to your kind advice, the apo form of NDM-1 was prepared by
dialysis against EDTA (lines 354–365). MST was performed again to detect the
binding affinities of apo-NDM-1 to Zn^{2+} , thanatin, and colistin in the revised
manuscript (lines 523–536). We measured a K_d value of $0.71 \pm 0.06 \mu M$ for the
interaction between NDM-1 and thanatin (Fig. 5c). Weak affinities to NDM-1 were
detected with K_d values of $7.36 \pm 0.45 \mu M$ for Zn^{2+} and $61.68 \pm 4.92 \mu M$ for colistin
(Supplementary Figs. 9a, b). These K_d values were lower than those detected with
NDM-1 loaded with Zn^{2+} , indicating higher affinity to apo-NDM-1.

**Question 3:** Metallo-beta-lactamases requires two metal ions to be active (Gonzalez
et al. (2012), Nature Chemical Biology, 8, 698). It is also likely that thanatin can
displace only one Zn (II) ion, inactivating the enzyme. This could be a plausible
explanation provided the authors measure the metal content of the enzyme and
measuring the affinities using the apo-protein.

**Answer:** We really appreciate your constructive advice. In the revised manuscript,
MST was used to redetect the binding affinities of apo-NDM-1 to Zn^{2+} , thanatin, and
colistin. The new results showed that thanatin had higher binding capacity to
apo-NDM-1 ($K_d = 0.71 \pm 0.06 \mu M$) than Zn^{2+} ($K_d = 7.36 \pm 0.45 \mu M$) (Fig. 5c,
Supplementary Fig. 9a). Hence, thanatin was probably capable to compete with Zn^{2+}
to bind to NDM-1. To further confirm whether thanatin causes the release of Zn^{2+} , we
conducted ICP-MS to measure the metal content of holo-NDM-1 after incubation with
or without thanatin. The ICP-MS results showed a loss of approximately half of the
zinc content in holo-NDM-1 after thanatin treatment (Fig. 5f).

**Question 4:** Lines 216-224 (Question 7). The authors clarify that they have used the
same chemically modified enzyme for all the binding experiments, allowing them to
compare the data. This is correct, but it does not address the issues I raised, since
these are artificial conditions and the relative affinities cannot be extrapolated to the
unmodified enzymes.

**Answer:** Thank you very much for your constructive comments on our study. As you

pointed out, lysines are present in the vicinity of the metal binding site, and the
NT-647-NHS dye reacts with the lysine residues to form highly stable dye–protein
conjugates. The binding affinities may be measured imprecisely due to this
modification. Accurately determining the binding affinity between thanatin and
NDM-1 is necessary. To accurately detect the binding affinities, at first, we considered
to conduct label-free MST assay, in which a fluorescent label is not needed. This
method uses Monolith NT. Label Free system to detect protein intrinsic fluorescence
dominated by the presence of tryptophan (λ excitation~280 nm, λ emission~350 nm).
However, the result of the broad-wavelength fluorescence scanning indicated that
thanatin shows two emission peaks at approximately 320 and 560 nm after excitation
at 280 nm. The intrinsic fluorescence of thanatin would affect the accuracy of the
label-free MST assay results. Thus, the label-free MST assay is unsuitable here.

The RED-tris-NTA second-generation dye (Nanotemper Technologies GmbH,
L018) can bind efficiently and site specifically to His-tag containing six or more
histidines. This dye is widely applied to detect binding affinities by MST assay due to
its minimal effect on the biochemical and physicochemical properties of the protein ^{2,3,}
⁴. Accordingly, we used this dye to label apo-NDM-1 with His-tag to avoid the
modification of the metal binding site.

**Question 5:** Many of the questions raised by this reviewer were answered by the
authors, but they did not incorporate these aspects in the manuscript text. For example,
indicating whether the in vitro experiments were performed with the holo or
apo-protein (Question 8) or with the processed soluble enzyme (Question 9) are
important for the reader. The expression “The NDM-1 gene” used in the Experimental
Section (line 306) is vague, and requires detail of the N and C-terminal residues of the
expressed proteins.

**Answer:** Thank you for your kind suggestions. We appreciate that the reviewer
pointed out the issues in the previous manuscript. We have modified these concerns
accordingly in the revised version.

1. We specified that apo-NDM-1 was used in the MST assay, and holo-NDM-1 was

used in the ICP-MS assay.

2. We modified the methods of expression and purification of NDM-1. We used
“*bla*_{NDM-1} gene” instead of “NDM-1 gene”. The detailed information is added,
wherein the signal peptide is deleted and the protein sequence is displayed (lines
322–331).

**Question 6:** Regarding the aspect of protein quantitation in Question 5, the authors do
not address our point regarding the use of different techniques. This problem could
have been sorted out by performing Western blots in supernatants and precipitates. We
also disagree with their assertion that loading controls are not used in Western blots.
For instance, in reference 10 (cited in line 194), GroEL is used as a loading control in
Figure 5.

**Answer:** According to your kind advice, the NDM-1 levels in the supernatant and
precipitates were detected by Western blot analysis with GroEL as a loading control in
the revised manuscript (Supplementary Fig. 8b). To clearly normalize the levels of
NDM-1, we investigated these two proteins on the same membrane. After detecting
the bands of NDM-1, the stripping buffer was used to remove the primary and
secondary antibodies from the membrane according to the manufacturer’s protocol
(Beijing Solarbio Science & Technology Co., Ltd., China). The stripped membrane
was then used to detect the levels of GroEL.

**Reviewer #2 (Remarks to the Author):**

**Reviewer’s comments:** The authors have tried very hard to address the comments
from the reviewers. They have revised incorrect wording as requested. The new data
regarding kinetic parameters support the assertion that thanatin inhibits NDM-1
competitively.

**Answer:** Thanks very much for your positive comments on our study. We really
appreciate it.

**Question 1:** The only remaining question deals with the measurement of. If a

suspension of cells is used to measure hydrolysis rates, part of the decrease in
absorbance must be due to the settling out of the cells during the reaction. Did the
authors run appropriate blanks without imipenem so that a correction factor could be
used in the calculations?

**Answer:** Thank you very much for your carefulness. In this assay, 50 μ L of the
bacterial cell suspension was diluted with 20 mM Tris-HCl (pH = 7.4) and transferred
to 96-well plates. The solution was added with 50 μ L of 200 μ M imipenem to measure
the hydrolytic activity of NDM-1 and 50 μ L of 20 mM Tris-HCl (pH = 7.4) was added
as blank. Data were corrected by subtracting the absorbance values obtained from the
blanks (Supplementary Figs. 8d, e). The detailed information of this assay has been
added in the revised manuscript to make the methods clear and precise (lines 482–
489).

**Question 2:** The authors should check the spelling in the manuscript and supplement.
There is at least one misspelling of "imipenem" as "imipeniem".

**Answer:** Thank you very much for your carefulness. We have checked the spelling in
the revised manuscript and Supplementary information. The writing of the manuscript
has also been polished by EssayStar Group (<http://essaystar.com/>), an English
language editing company.

**Reviewer #3 (Remarks to the Author):**

No further comments for author.

**References**

- 1. Gonzalez LJ, Bahr G, Nakashige TG, Nolan EM, Bonomo RA, Vila AJ.
Membrane anchoring stabilizes and favors secretion of New Delhi
metallo-beta-lactamase. *Nat Chem Biol* 2016, 12(7): 516-522.
2. Bartoschik T, Galinec S, Kleusch C, Walkiewicz K, Breitsprecher D, Weigert
S, et al. Near-native, site-specific and purification-free protein labeling for
quantitative protein interaction analysis by MicroScale Thermophoresis. *Sci*

Rep 2018, 8(1): 4977.

3.

Lata S, Gavutis M, Tampe R, Piehler J. Specific and stable fluorescence labeling of histidine-tagged proteins for dissecting multi-protein complex formation. *J Am Chem Soc* 2006, 128(7): 2365-2372.

4.

<https://resources.nanotempertech.com/application-notes/one-step-purification-free-and-site-specific-labeling-of-polyhistidine-tagged-proteins-for-mst>

REVIEWERS' COMMENTS:

Reviewer #1 (Remarks to the Author):

The authors have adequately addressed with additional experiments and thoughtful discussion the issues of concern. This has resulted in a considerably improved manuscript.

RESPONSE TO REVIEWERS' COMMENTS

Reviewer #1 (Remarks to the Author):

The authors have adequately addressed with additional experiments and thoughtful discussion the issues of concern. This has resulted in a considerably improved manuscript.

Answer: Thanks very much for your positive comments. We appreciate that the reviewer has provided many constructive suggestions to help us improve our study.